# Generalizable Reasoning through Compositional Energy Minimization

**Alexandru Oarga**
University of Barcelona

**Yilun Du**
Harvard University

## Abstract

Generalization is a key challenge in machine learning, specifically in reasoning tasks, where models are expected to solve problems more complex than those encountered during training. Existing approaches typically train reasoning models in an end-to-end fashion, directly mapping input instances to solutions. While this allows models to learn useful heuristics from data, it often results in limited generalization beyond the training distribution. In this work, we propose a novel approach to reasoning generalization by learning energy landscapes over the solution spaces of smaller, more tractable subproblems. At test time, we construct a global energy landscape for a given problem by combining the energy functions of multiple subproblems. This compositional approach enables the incorporation of additional constraints during inference, allowing the construction of energy landscapes for problems of increasing difficulty. To improve the sample quality from this newly constructed energy landscape, we introduce Parallel Energy Minimization (PEM). We evaluate our approach on a wide set of reasoning problems. Our method outperforms existing state-of-the-art methods, demonstrating its ability to generalize to larger and more complex problems. Project website can be found at: https://alexoarga.github.io/compositional_reasoning/

## 1 Introduction

Being able to solve complex reasoning problems, such as logical reasoning, combinatorial puzzles and symbolic manipulation, is one of the key challenges in machine learning. This is particularly interesting because it requires models to go beyond pattern recognition. For a model to successfully perform reasoning tasks, it is expected to be able to generalize to unseen distributions during test time. That is, they are expected to solve not only problems similar to those encountered during training, but also to be able to generalize to novel conditions and distributions [9].

The standard paradigm in machine learning for solving reasoning tasks is to train models end-to-end to map inputs to outputs. During training, models are exposed to a large number of solutions and learn statistical heuristics that allow them to solve similar problems. This contrasts with human reasoning, where we first learn the rules and constraints governing a problem, and then apply them in a compositional manner to arrive at a solution. Notably, humans are able to solve such problems without having seen an exact solution before. Moreover, when prompted with harder tasks, we can invest more time, effectively searching for a solution, rather than relying on heuristics [1, 37].

In this work, we present an approach to reasoning where the overall process is cast as an optimization problem [22]. Specifically, we learn an energy function $E_\theta(\boldsymbol{x}, \boldsymbol{y})$ across all possible solutions $\boldsymbol{y}$ of the problem, where $\boldsymbol{x}$ are the given conditions of the problem. This energy function is learned such that valid solutions are assigned lower energy, while invalid solutions receive higher energy.

---

Corresponding author: ydu@seas.harvard.edu

39th Conference on Neural Information Processing Systems (NeurIPS 2025).

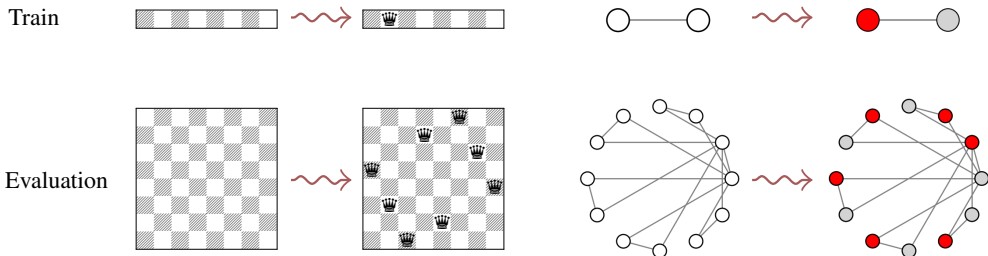

Figure 1: **Compositional Generalizable Reasoning.** We formulate reasoning as optimization problem with inputs $x$ and solutions $y$. By combining multiple optimization objectives, we can generalize to larger problem instances (bottom) than those seen during training (top). This enables us to solve a complex instance of N-queens (left) or a more complex instance of graph coloring (right).

Reasoning corresponds to minimizing the energy function to find low-energy solutions. Harder problems can be solved by spending more time reasoning and minimizing the optimization objective.

To solve more complex problems than those seen during training (see Figure 1), we can jointly reason and minimize the sum of several optimization objectives at the same time. For instance, in logical reasoning, we can learn the energy landscape of individual clauses, and then optimize over multiple clauses simultaneously. This composition of energy objectives enables the model to find assignments that satisfy all clauses simultaneously, enabling it to solve larger, more complex problems. However, combining multiple objectives makes the landscape increasingly complex and introduces local minima, making it hard to optimize. To address this, we propose a parallel strategy where we use a system of particles for optimization. In this setup, we leverage the energy function as a resampling mechanism to improve the quality of the samples and avoid local minima that attract particles. This approach improves exploration and ultimately makes optimization more effective.

We illustrate the applicability of our approach across a set of difficult reasoning problems, including the N-Queens, 3-SAT and the Graph Coloring. We compare against domain-specific state of the art combinatorial optimization models, and show that our approach outperforms them in terms of solution quality and generalization to larger and more complex problems. We further show that, by adjusting the computational budget, we can adapt the model to solve more complex problems. Finally, ablation studies show that our training strategy leads to improved results and that our sampling strategy is able to produce better quality solutions than existing samplers.

Overall, the contributions of this work are threefold. First, we propose a compositional approach to reasoning generalization, where we combine energy landscapes during inference to solve more complex problems. Second, we introduce a new sampling strategy, Parallel Energy Minimization (PEM), a particle-based optimization strategy that enables us to effectively optimize composed energy functions to solve hard reasoning tasks. Finally, we illustrate the efficacy of our approach empirically across a wide set of reasoning tasks, outperforming many existing combinatorial optimization approaches in generalization.

## 2  Related work

**Reasoning as Optimization.** Reasoning includes multiple cognitive processes, such as logical inference, decision-making, planning, and scheduling. Many of these can be formulated as optimization problems, where the goal is to find variable assignments that minimize an objective function under certain constraints. Prior works have integrated logical reasoning into neural networks through differentiable relaxations of SAT [26] and MAXSAT solvers [64], differentiable theorem proving [53, 47], probabilistic logic [45], differentiable logic rules [59], and gradient-based methods for discrete distributions [49]. Other approaches incorporate continuous optimization directly into models via differentiable convex [2], quadratic [5] or integer solvers [61]. These methods, however, often target specific domains or rely on strong assumptions.

Another line formulates reasoning using general-purpose optimization frameworks. For instance, [54, 15] simulate physical dynamics using energy minimization. Latent space optimization has been used in variational methods for molecule generation [28] and the Traveling Salesman Problem (TSP) [35]. More related to our work, [22] learns energy functions backpropagating through optimization steps or with diffusion-based losses [23]. Nonetheless, most of these approaches adopt end-to-

end methods to learn reasoning tasks, limiting their generalization ability. We instead propose a compositional strategy, combining energy landscapes learned on subproblems to tackle larger tasks.

Finally, recent approaches focus on learning-based methods for Combinatorial Optimization (CO), which aim to reduce the computational cost by generating near-optimal solutions. Graph Neural Networks (GNNs) are the standard in this domain due to their ability to represent variable-constraint relations. Recent works have employed GNNs to directly predict solutions [10, 36, 55], learning discrete diffusion over graphs [60, 42], using reinforcement learning [8, 3] or learning Markov processes [72]. However, it is well known that GNNs struggle out of distribution [67, 27] and require large, diverse datasets. Our approach leverages the compositional nature of reasoning problems by producing more generalizable energy landscapes combining multiple energy objectives.

**Reasoning as Iterative Computation.** Some strategies use neural networks to iteratively refine solutions to reasoning problems. This motivation is drawn from optimization solvers, which operate with iterative updates. Within this category, we can identify three main directions: (1) methods that incorporate explicit program representations [30, 52, 12, 70, 48], (2) works based on recurrent neural networks [29, 38, 13, 69, 18, 56, 70], and (3) techniques that approximate solutions via iterative refinement [58, 7, 43, 42, 64, 44]. In our work, we cast reasoning problems as optimization problems, hence we use optimization algorithms as refinement steps for solution search.

**Energy-Based Models and Diffusion Models.** Our work is closely related to Energy-Based Models (EBMs) [32, 40, 24]. Most of the work in this field has focused on learning probabilistic models over data [24, 50, 21, 6, 66, 17]. In contrast, we train an EBM for solving reasoning tasks by performing optimization over the learned energy landscape.

## 3 Method

### 3.1 Reasoning as Energy Minimization

Let $\mathcal{D} = \{X, Y\}$ be a dataset of reasoning problems with inputs $\boldsymbol{x} \in \mathbb{R}^O$ and solutions $\boldsymbol{y} \in \mathbb{R}^M$. We wish to find an operator $f(\cdot)$ that can generalize to test problems $f(\boldsymbol{x}')$ where $\boldsymbol{x}' \in \mathbb{R}^{O'}$, is potentially larger and more complex than $\boldsymbol{x}$. Let $E_\theta(\boldsymbol{x}, \boldsymbol{y}) : \mathbb{R}^O \times \mathbb{R}^M \to \mathbb{R}$, be an EBM defined across all possible solutions $\boldsymbol{y}$ given $\boldsymbol{x}$, such that ground-truth solutions $\boldsymbol{y}$ are assigned lower energy. Finding a solution to the reasoning problem corresponds to finding an assignment $\hat{\boldsymbol{y}}$ such that:

$$\hat{\boldsymbol{y}} = \arg\min_{\boldsymbol{y}} E_\theta(\boldsymbol{x}, \boldsymbol{y}) \tag{1}$$

To find the solution $\hat{\boldsymbol{y}}$, one can use gradient descent:

$$\boldsymbol{y}^t = \boldsymbol{y}^{t-1} - \lambda \nabla_{\boldsymbol{y}} E_\theta(\boldsymbol{x}, \boldsymbol{y}^{t-1}) \tag{2}$$

where $\lambda$ is the step size, and $\boldsymbol{y}^0$ is the initial solution drawn from a fixed noise distribution (e.g. Gaussian). The resulting solution $\boldsymbol{y}^T$ is found after $T$ iterations of the above update.

**Diffusion Energy-Based Models.** The effective training of EBMs is a challenging task, and currently, many approaches exist in the literature for this purpose [24, 11, 20]. Previous works trained EBMs by backpropagating the gradient through T generative steps [22]. However, this could lead to instabilities in the training and high computational cost of backpropagation.

In this work, we propose instead to use the denoising diffusion training objective introduced in [20]. Specifically, we train the gradient of the EBM to match the noise distribution at each timestep $t$. Formally, given a truth label $\boldsymbol{y}$ from the dataset, and a gaussian corrupted label $\boldsymbol{y}^*$, where $\boldsymbol{y}^* = \sqrt{1 - \sigma_t}\boldsymbol{y} + \sigma_t\epsilon$ and $\epsilon \sim \mathcal{N}(0, I)$ we can define the diffusion objective as:

$$\mathcal{L}_{\text{MSE}}(\theta) = \mathbb{E}_{\boldsymbol{y}, \epsilon \sim \mathcal{N}(0, I)} \left[ \|\epsilon + \sigma_t \nabla_{\boldsymbol{y}} E_\theta(\boldsymbol{y}^*, t)\|^2 \right] \tag{3}$$

with $E_\theta(\boldsymbol{y}^*, t)$ being a explicitly defined scalar function[0], and $\sigma_t$ a sequence of fixed noise schedules.

This formulation allows us to supervise the gradient of the energy function at each optimization step $t$, avoiding the need to backpropagate through a sequence of $T$ steps. As a result, we learn an energy gradient that transforms a noisy input into the target distribution, through a series of optimization steps. To generate outputs, we can then use, for example, the update rule given in (2).

---

[0]Empirically, in this work we use the same function as in the original paper, this is, $E_\theta(x_t, t) = \|s_\theta(x_t, t)\|^2$, where $s_\theta(x_t, t)$ is a vector-output neural network.

**Shaping the Energy landscape.** The training objective presented in Eq. (7) does not guarantee that the target label $\boldsymbol{y}$ is assigned the energy minima of the energy landscape. In this work, to enforce that the energy minima align to the ground-truth label, and to enhance regions of the landscape not covered by the diffusion-based training, we follow the approach of [23], and introduce an additional contrastive loss function to shape the energy landscape.

This contrastive loss guides the energy function by comparing noise-corrupted labels of given pairs of positive and negative samples. Formally, the objective at step $t$ is formulated as:

$$\mathcal{L}_{CL}(\theta) = -\log\left(\frac{e^{E^+}}{e^{E^+} + \sum e^{E^-}}\right) \tag{4}$$

where $E^+ = E_\theta(\tilde{\boldsymbol{y}}^+, t)$ and $E^- = E_\theta(\tilde{\boldsymbol{y}}^-, t)$, with $\tilde{\boldsymbol{y}}^+$ and $\tilde{\boldsymbol{y}}^-$ being positive and negative noise corrupted samples respectively, this is, $\tilde{\boldsymbol{y}}^+ = \sqrt{1-\sigma_t}\boldsymbol{y}^+ + \sigma_t\epsilon$ and $\tilde{\boldsymbol{y}}^- = \sqrt{1-\sigma_t}\boldsymbol{y}^- + \sigma_t\epsilon$.

## 3.2 Compositional Reasoning

We wish to construct a reasoning framework that can generalize to complex problems that are much harder than those seen at training time, consisting of a significantly greater number of constraints. To construct an effective energy function to tackle such problems, we propose to decompose the energy function into smaller ones that are defined over tractable subproblems that have been seen before. These subproblems are then simpler to handle and represent with energy functions compared to trying to solve the original problem. In particular, we propose decomposing a full reasoning problem $\boldsymbol{x}$ into simpler subproblems $\boldsymbol{x} = \{\boldsymbol{x_1}, \ldots, \boldsymbol{x_N}\}$, such that finding a solution $\boldsymbol{y_i}$ to each subproblem $\boldsymbol{x_i}$ solves the original problem $\boldsymbol{x}$.

Given this decomposition, let $E_\theta^k(\boldsymbol{x}, \boldsymbol{y})$ be an EBM of the $k$-th subproblem, where $\boldsymbol{y_k}$ is assigned the lowest energy when it is a solution to subproblem $\boldsymbol{x_k}$. A complete solution $\hat{\boldsymbol{y}}$ to the original problem $\boldsymbol{x}$ is obtained by solving all subproblems simultaneously, formally optimizing the composition of each energy function:

$$\hat{\boldsymbol{y}} = \arg\min_{\boldsymbol{y}} \sum_{k=1}^{N} E_\theta^k(\boldsymbol{x_k}, \boldsymbol{y_k}) \tag{5}$$

where $\hat{\boldsymbol{y}}$ can be found as in (2). We illustrate how to effectively optimize these objectives next.

## 3.3 Improving Sampling with Parallel Energy Minimization

Optimization over EBMs can be done through approximate methods such as Markov Chain Monte Carlo (MCMC). MCMC simulates a Markov chain, starting from an initial state $\boldsymbol{y_0}$, drawn from a noise distribution, with subsequent samples generated from a transition distribution. A common approach to MCMC sampling in EBMs is Unadjusted Langevin Dynamics (ULA) [24, 50], which is defined as:

$$\boldsymbol{y}^t = \boldsymbol{y}^{t-1} - \lambda\nabla_{\boldsymbol{y}}E_\theta(\boldsymbol{x}, \boldsymbol{y}^{t-1}) + \sqrt{2}\lambda\xi, \quad \xi \sim \mathcal{N}(0,1) \tag{6}$$

where $\lambda$ is the step size of the optimization method. This essentially corresponds to performing gradient descent on the energy function with some added noise.

However, such a noisy optimization procedure will often become stuck in local minima, which are especially prevalent in composed energy landscapes such as Equation 5. In MCMC, a common technique to more effectively sample from such difficult probability distributions is Sequential Monte Carlo (SMC) [19], where a set of parallel particles is maintained and evolved over iterations of sampling to help prevent premature convergence to local minima.

Inspired by this insight, we propose Parallel Energy Minimization (PEM), a parallel optimization procedure for optimizing composed energy landscapes. We initialize and optimize a parallel set of $P$ particles across the T steps of optimization as presented in Algorithm 1 and illustrated in Figure 2. At each optimization step, PEM resamples particles based on their energy values, allowing local minima to be discarded. To further help particles cover the entire landscape of solutions, we add a predefined amount of noise to each particle each time they are resampled.

**Algorithm 1** Parallel Energy Minimization (PEM)

---

**Input:** $T$ optimization steps, $P$ particles

Given a set of particles $\{y_i^{(T)}\}_{i=1}^P, y_i^{(T)} \sim \mathcal{N}(0,1)$
**for** timestep $t$ in $T, \dots, 1$ **do**
  ▷ *Importance evaluation*
  $w^{(t)} \leftarrow \text{softmax}(-E_\theta(y^{(t)}, t))$
  ▷ *Selection*
  Resample $y^{(t)}$ based on weights $w^{(t)}$
  ▷ *Resampling*
  $\tilde{y}^{(t)} \leftarrow y^{(t)} + \sigma_t \xi \quad \xi \sim \mathcal{N}(0,1)$
  ▷ *Optimize solutions with gradient*
  $y^{(t-1)} \leftarrow \tilde{y}^{(t)} + \sigma_t \nabla E_\theta(\tilde{y}^{(t)}, t)$
**end for**
**return** $x^{(0)}$

---

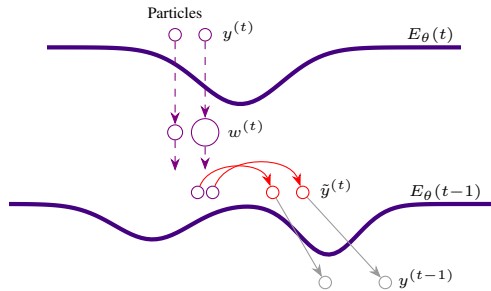

Figure 2: **PEM Sampling.** At timestep $t$, particles $y^{(t)}$ are first resampled using weights $w^{(t)}$ derived from $E_\theta(y^{(t)}, t)$. Next, scheduled Gaussian noise is added to obtain a new set of resampled particles. Finally, the particles of the next timestep $t-1$ are generated optimizing the gradient of the energy function at time $t$.

## 3.4 Refinement of the Energy Landscape

Composing multiple energy objectives together can effectively generate complex energy landscapes suitable for solving larger problems. However, as the number of objectives increases, the energy landscape becomes increasingly complex, which can lead to inaccuracies in the overall energy function. In particular, minima might appear in the function that incorrectly assigns lower energy to invalid solutions.

To mitigate this issue, we propose a refinement strategy for the composed landscape. Given a set of $N$ energy functions $\{E_\theta^k(\boldsymbol{x}_k, \boldsymbol{y}_k)\}_{k=1}^N$, each trained on a subproblem $\boldsymbol{x}^k$, we refine the composed energy function using ground-truth solutions $\boldsymbol{y}$. The resulting training objective is defined as:

$$\mathcal{L}_{\text{MSE}}(\theta) = \mathbb{E}_{\boldsymbol{y}, \mathcal{N}(\epsilon, 0, I)} \| \epsilon + \sigma_t \nabla_{\boldsymbol{y}} \sum_{k=1}^N E_\theta^k(\boldsymbol{y}^*, t) \|^2 \tag{7}$$

having $\boldsymbol{y}^* = \sqrt{1 - \sigma_t}\boldsymbol{y} + \sigma_t \epsilon$ and $\epsilon \sim \mathcal{N}(0, I)$. This refinement helps align energy minima with valid solutions, correcting inaccuracies and improving robustness.

# 4 Experiments

## 4.1 N-Queens Problem

**Setup.** The N-queens problem involves placing $N$ queens on an $N \times N$ chessboard such that no two queens threaten each other, meaning no two queens can be placed in the same row, column or diagonal. We evaluate how well different methods can generate valid solutions to the problem. During training we use only one single instance of the N-queens problem for a given value $N$. In our approach, we use the $N$ rows of a single instance to train, and then compose this model row-wise, column-wise, and diagonal-wise to form a 2D chessboard. That is, we train a model to generate a valid row and then reuse it simultaneously for rows, columns and diagonals (see Figure 3).

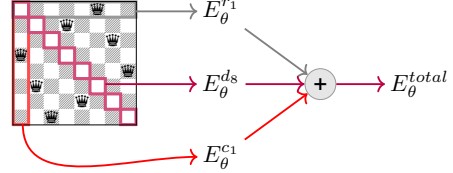

Figure 3: **N-Queens Problem Composition.** To compose a row model to solve the N-queens problem, we add the energy of each row $i$ ($E_\theta^{ri}$), each column $j$ ($E_\theta^{cj}$), each diagonal $k$ ($E_\theta^{dk}$) of the chessboard. We then sample from the resulting energy function $E_\theta^{total}$ to generate valid solutions.

**Baselines.** We compare against existing baselines for neural combinatorial optimization solvers, for which the N-queens problem is represented as a graph, and the solution corresponds to a Maximum Independent Set (MIS). As baselines we include: a reinforcement learning approach, where the model learns to defer harder nodes when solving the problem (*LWD*, [3]), an unsupervised method, where the model learns a Markov decision process over graphs (*GFlowNets*, [72]), and a supervised categorical diffusion solver (*DIFUSCO*, [60]). Furthermore, we also compare against previous state

| Model | Type | Correct Instances ↑ | Size ↑ |
|---|---|---|---|
| LWD | RL + S | 22 | $7.1000 \pm 0.5744$ |
| GFlowNets | UL + S | 14 | $6.9293 \pm 0.5904$ |
| DIFUSCO ($T$=50) | SL + S | 17 | $6.9400 \pm 0.6452$ |
| Fast T2T ($T_S$=1, $T_G$=1) | SL + S | 21 | $6.8200 \pm 0.8761$ |
| Fast T2T ($T_S$=1, $T_G$=1) | SL + GS | 12 | $6.7000 \pm 0.7141$ |
| Fast T2T ($T_S$=5, $T_G$=5) | SL + S | 20 | $7.0600 \pm 0.5800$ |
| Fast T2T ($T_S$=5, $T_G$=5) | SL + GS | 41 | $7.3800 \pm 0.5436$ |
| EBM ($P$=1024) (Ours) | SL + PEM | **97** | $\mathbf{7.9699 \pm 0.1714}$ |

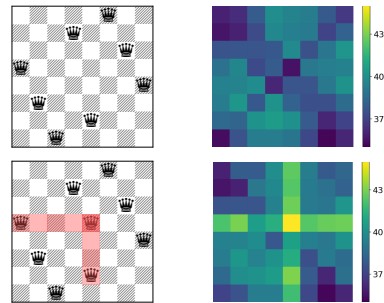

Table 1: **8-Queens Problem Evaluation.** We compare the performance against state-of-the-art combinatorial optimization models on the 8-queens solution generation task. All the models were trained with 1 single instance of the 8-queens problem. We sampled 100 8-queens solutions. IR: Iterative Refinement, BP: Belief Propagation, TS: Tree Search, S: Sampling, GS: Guided Sampling,

Figure 4: **Energy Map Visualization.** Correct solutions (top, left) are assigned low energy (top, right) and incorrect solutions (bottom, left) are assigned higher energy (bottom, right). Energy at each position is the sum of the row, column, and diagonal energy.

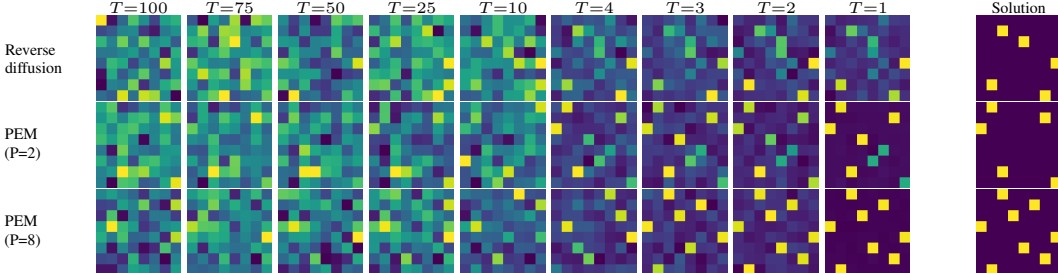

Figure 5: **Optimized Samples Across Timesteps.** Generated samples where yellow squares represent queens placed in the chessboard. Reverse diffusion fails to find valid solutions (top). Increasing the number of particles with PEM leads from invalid solutions (middle) to valid solutions (bottom).

of the art combinatorial optimization models (*Fast T2T* [42]), with different inference steps $T_s$ and gradient search steps $T_g$. For the latter, we also compared against the guided sampling version, where a penalty function is added for the MIS problem to guide denoising.

For all the methods evaluated, we report the number of correct instances of the problem found, and the average number of queens placed in the chessboard. We follow previous works approach to decode solutions, where, given a model heatmap of the chessboard, we perform greedy decoding by sequentially placing queens in the board until a conflict is found.

**Quantitative Results.** We report the comparison of our approach with the previous baselines in Table 1. For all the methods reported, we sampled 100 different solutions. In this table, we can see that our approach is able to generate nearly all perfect solutions to the problem. Furthermore, our method significantly outperforms the previous state-of-the-art solvers. Out of 100 generated samples, 97 are valid 8-queens solutions, while state-of-the-art methods are able to find 41 correct instances at most.

**Qualitative Results.** In Figure 5 we visualize the sampling process of our approach using reverse diffusion and PEM. We can observe that PEM is able to generate much better quality samples than reverse diffusion. Additionally, more particles leads to the generation of valid solutions. We include an example of parallel sampling with 8 particles in Appendix B. In Figure 4 we can see that a higher energy is assigned to rows and columns where the constraints are violated.

**Performance with Increased Computation.** In Table 2 we report the performance of our approach on the 8-queens problem with increasing number of particles during sampling. We show that increasing the number of particles significantly improves the quality of the generated samples and, as a consequence, a larger number of correct instances are found. In Figure 6, we visualize the performance of our model with different number of particles on different complexity levels of the problem. We can see that by adjusting the number of particles, we can adapt the ability of our model to solve more difficult problems.

| Num. Particles | Correct Instances ↑ | Size ↑ |
|---|---|---|
| 8 | 9 | $6.6599 \pm 0.7550$ |
| 64 | 34 | $7.2500 \pm 0.6092$ |
| 128 | 87 | $7.8800 \pm 0.3265$ |
| 256 | 89 | $7.9000 \pm 0.3015$ |
| 512 | 92 | $7.9400 \pm 0.2386$ |
| 1024 | 97 | $7.9699 \pm 0.1714$ |

Table 2: **Number of Particles vs Correct Instances.** We sampled 100 solutions from the 8-queens problem. Increasing the number of particles with PEM significantly improves the number of correct instances.

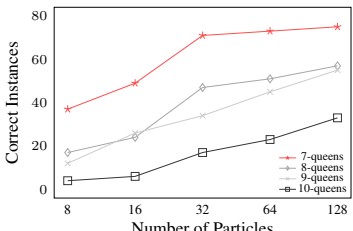

Figure 6: **Number of Particles vs Correct Instances Across Problem Difficulties.** We sampled 100 solutions each from N-queens problems of increasing difficulty (7 to 10 queens). In all cases, a higher number of particles results in more correct instances.

| Sampler | Correct Instances ↑ | Size ↑ |
|---|---|---|
| Reverse Diffusion | 12 | $2.6400 \pm 0.7722$ |
| ULA | 5 | $2.5200 \pm 0.6432$ |
| MALA | 8 | $2.6700 \pm 0.6824$ |
| UHMC | 9 | $2.6700 \pm 0.6971$ |
| HMC | 11 | $2.6900 \pm 0.7204$ |
| PEM ($P$=8) | 99 | $3.9900 \pm 0.1000$ |

Table 3: **Sampler Ablation.** Ablations proposed for different samplers. We sampled 100 solutions from the 4-queens problem. Compared to other samplers, PEM is able to consistently produce accurate solutions.

| Diffusion Loss | Contrastive Loss | Correct Instances ↑ | Satisfied Clauses ↑ |
|---|---|---|---|
| No | Yes | 0 | $1.3200 \pm 0.5482$ |
| Yes | No | 6 | $6.6799 \pm 0.7089$ |
| Yes | Yes | 97 | $7.9699 \pm 0.1714$ |

Table 4: **Loss Ablation.** Ablations proposed for the loss function on the performance on the 8-queens problem. We sampled 100 solutions from the 8-queens problem. A combination of both a diffusion and contrastive loss to shape the landscape produces the best results on the task. In all cases we sampled using PEM ($P$=1024).

**Ablation Study.** In Table 3, we compare the performance with different methods proposed for EBM sampling, including Unadjusted Langevin Dynamics (ULA), Metropolis Adjusted Langevin Dynamics (MALA), Unadjusted Hamiltonian Monte Carlo (UHMC) and Hamiltonian Monte Carlo (HMC). We show that our approach substantially outperforms existing samplers in the 4-queens task. In Table 4, we compare three models trained with different loss functions. We show that the combination of the diffusion and contrastive losses produces improved results on the task.

## 4.2 SAT Problem

**Setup.** In this section we evaluate the performance of our approach on the Boolean satisfiability problem (SAT), well known to be an NP-complete problem. The 3-SAT problem is a binary decision problem where a Boolean formula is given in Conjunctive Normal Form (CNF), with each clause having exactly 3 literals. The task is to find a truth assignment to the variables (true or false) such that the formula evaluates to true. For training, we generated random 3-SAT instances with number of variables within $[10, 20]$. The number of clauses was set to be in phase transition, that is, it was set to be $4.258 \times n$, where $n$ is the number of variables [57]. For our approach, we train a model to generate a satisfiable assignment to only one individual clause of the 3-SAT problem. We then compose the model to generate a solution to the entire problem. This enables the generalization to an arbitrary number of clauses. We evaluate using the SATLIB benchmark [34]. For a distribution similar to the training one, we used 100 instances with 20 variables and 91 clauses. For a larger distribution, we used 100 instances with 50 variables and 218 clauses.

**Baselines.** We compare against existing baselines for neural SAT solvers, including: the seminal neural SAT solver, where the solution is iteratively refined with increasing number of steps (NeuroSAT [58]), and the state-of-the-art neural solver based on belief-propagation (NSNet [43]). In both cases, we use different number of steps $T$ for solution refinement. When feasible, we also compare with combinatorial optimization models by encoding the 3-SAT problem as a graph.

**Quantitative Results.** In Table 5 we find that our method significantly outperforms the previous state-of-the-art neural SAT solvers, and is able to find a larger number of correct instances of the problem. In the similar distribution 91 instances are solved compared to 58 instances solved by NSNet. In the larger distribution our method still outperforms the previous other methods with 43 correct instances, with NSNet solving 37 correct instances.

| Model | Type | Similar Distribution | | Larger Distribution | |
|---|---|---|---|---|---|
| | | **Correct Instances** ↑ | **Satisfied Clauses** ↑ | **Correct Instances** ↑ | **Satisfied Clauses** ↑ |
| GCN | SL | 5 | $0.9617 \pm 0.0264$ | 0 | $0.9569 \pm 0.0203$ |
| DGL | SL + TS | 10 | $0.9520 \pm 0.0330$ | 0 | $0.8705 \pm 0.0405$ |
| DIFUSCO ($T = 50$) | SL + S | 6 | $0.9734 \pm 0.0156$ | 0 | $0.9738 \pm 0.0156$ |
| Fast T2T ($T_S{=}1, T_G{=}1$) | SL + S | 23 | $0.9749 \pm 0.0210$ | 4 | $0.9751 \pm 0.0141$ |
| Fast T2T ($T_S{=}5, T_G{=}5$) | SL + S | 22 | $0.9760 \pm 0.0273$ | 20 | $0.9734 \pm 0.0159$ |
| NeuroSAT ($T{=}50$) | SL + IR | 6 | $0.9661 \pm 0.0185$ | 0 | $0.9651 \pm 0.0110$ |
| NeuroSAT ($T{=}500$) | SL + IR | 8 | $0.9742 \pm 0.0154$ | 0 | $0.9697 \pm 0.0111$ |
| NSNet ($T{=}50$) | SL + BP | 58 | $0.9845 \pm 0.0272$ | 34 | $0.9817 \pm 0.0237$ |
| NSNet ($T{=}500$) | SL + BP | 58 | $0.9856 \pm 0.0266$ | 37 | $0.9846 \pm 0.0205$ |
| EBM ($P{=}1024$) (Ours) | SL + PEM | **91** | $\mathbf{0.9985 \pm 0.0048}$ | **43** | $\mathbf{0.9963 \pm 0.0046}$ |

Table 5: **3-SAT Problem Evaluation.** We compare the performance against the state-of-the-art combinatorial optimization models and neural SAT solvers on the 3-SAT task. Models are evaluated on a distribution similar to the training distribution and a larger distribution. Similar distribution has 100 instances with 20 variables and 91 clauses, while larger distribution has 100 instances with 50 variables and 218 clauses. Our approach outperforms existing methods. IR: Iterative Refinement, BP: Belief Propagation, TS: Tree Search, S: Sampling,

| Num. Particles | Similar Distribution | | Larger Distribution | |
|---|---|---|---|---|
| | **Correct Instances** ↑ | **Satisfied Clauses** ↑ | **Correct Instances** ↑ | **Satisfied Clauses** ↑ |
| 8 | 30 | 0.9874 | 2 | 0.9910 |
| 64 | 70 | 0.9962 | 7 | 0.9910 |
| 128 | 78 | 0.9975 | 18 | 0.9927 |
| 1024 | 91 | 0.9985 | 43 | 0.9963 |

Table 6: **Number of Particles vs 3-SAT Performance.** We compare the evaluation performance of our approach on the 3-SAT problem with increasing number of particles. Increasing the number of particles substantially improves the generalization performance of the model.

| Finetuning | Correct Instances ↑ | Satisfied Clauses ↑ |
|---|---|---|
| No | 57 | $0.9951 \pm 0.0068$ |
| Yes | 91 | $0.9985 \pm 0.0048$ |

Table 7: **Fine-tuning Ablation.** Ablations proposed for the finetuning of the model on the performance on the 3-SAT problem. We show that finetuning the composed model with complete instances leads to better performance. In all cases we sampled using PEM (P=1024).

**Qualitative Results.** In Appendix B, we present additional qualitative results for 3-SAT, where we show that unsatisfied clauses are assigned higher energy, while satisfied clauses are assigned lower.

**Performance with Increased Computation.** We assess in Table 6, the impact of the number of particles on the performance on the 3-SAT problem. We show that increasing the number of particles improves the results on both the similar and larger distributions. By formulating the problem as an energy minimization problem, we can adjust the number of particles to adapt to the difficulty of the task.

**Ablation Study.** In Appendix C we include ablation for the sampling procedure, showing that PEM outperforms existing methods, as well as ablations on the training loss. Moreover, in Table 7 we report that finetuning the composed model improves the overall performance.

## 4.3 Graph Coloring

**Setup.** In this section, we evaluate our approach on the graphical problem of graph coloring. Given a graph instance, the task is to assign a color to each node in the graph such that no two adjacent nodes share the same color using at most $k$ colors. This problem is known to be NP-complete. The chromatic number $\chi$ of a graph is the minimum number of colors needed to color the graph. To train baselines, we followed the same approach as in [41], and generated random graphs with number of nodes within $[20, 40]$, density within $[0.01, 0.5]$, and chromatic number $\chi$ within $[3, 8]$. For our approach, we train a model to generate a valid coloring of an individual edge given a set of colors. We then compose the model for all the edges of the graph to generate a valid coloring solution. To train our model we generate random pairs of different colors. For evaluation, we use graphs from the well-known COLOR benchmark[1]. Additionally, we also evaluate on random graph instances generated following different graph distributions, namely: Erdos-Renyi [25], Holme-Kim [33], and random regular expander graphs [4]. For each distribution, we generate smaller graphs with nodes within $[20, 40]$ and larger graphs with nodes within $[80, 100]$. Moreover, we also evaluate on densely connected regular graphs such as Paley graphs [51] and complete graphs. For all the methods, we

---
[1] `https://mat.tepper.cmu.edu/COLOR02/`

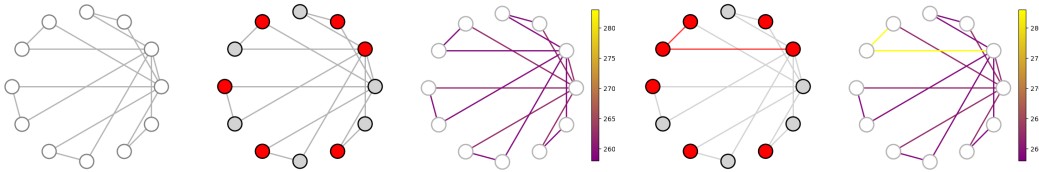

(a) Original Graph    (b) Correct Solution    (c) Correct Solution Energy    (d) Incorrect Solution    (e) Incorrect Solution Energy

Figure 7: **Qualitative Illustration of Energy Maps.** We show a graph instance (a) along with a valid solution in (b). In (c), we visualize the energy map of a correct solution, where we show the energy of each edge individually. We present an incorrect solution with two conflicting edges in (d) and the corresponding energy for each edge in (e). As expected, a higher energy is assigned to conflicting edges.

| Distribution | $\mathcal{V}$ | $\mathcal{E}$ | $d$ | $\chi$ | GCN | GAT | XLVIN | GNN-GCP | EBM (Ours) (P=128) |
|---|---|---|---|---|---|---|---|---|---|
| Erdos Renyi | [20, 39] | [29,76] | 0.12 | [3, 4] | 46.80 ± 20.47 | 34.00 ± 11.55 | 25.00 ± 7.81 | 15.20 ± 4.32 | 8.60 ± 4.82 |
| | [81, 99] | [193, 225] | 0.05 | [3, 4] | 151.60 ± 12.09 | 130.20 ± 11.47 | 93.80 ± 31.12 | 53.80 ± 8.34 | 29.20 ± 8.05 |
| Holme Kim | [22, 34] | [56, 92] | 0.26 | [4, 4] | 74.00 ± 14.74 | 51.20 ± 10.03 | 29.00 ± 7.75 | 13.20 ± 7.46 | 10.60 ± 2.70 |
| | [86, 100] | [398, 469] | 0.10 | [5, 6] | 408.00 ± 26.40 | 253.20 ± 50.71 | 182.60 ± 24.73 | 55.20 ± 12.63 | 59.00 ± 3.74 |
| Regular Expander | [21, 40] | [63, 120] | 0.22 | [4, 4] | 87.60 ± 22.58 | 58.60 ± 12.44 | 29.00 ± 7.75 | 15.40 ± 6.65 | 11.00 ± 4.89 |
| | [86, 100] | [184, 200] | 0.23 | [3, 3] | 144.80 ± 6.90 | 118.80 ± 12.59 | 112.60 ± 10.97 | 141.60 ± 69.47 | 37.20 ± 4.71 |
| Paley | [19, 37] | [171, 465] | 0.80 | [6, 10] | 285.00 ± 117.70 | 239.20 ± 159.43 | 151.80 ± 92.13 | 91.20 ± 63.14 | 34.80 ± 20.27 |
| Complete | [8, 12] | [36, 66] | 1.00 | [8, 12] | 46.00 ± 15.04 | 46.00 ± 15.04 | 34.80 ± 16.42 | 30.00 ± 2.54 | 3.40 ± 1.14 |

Table 8: **Graph Coloring Evaluation.** We compare the performance against canonical GNNs and GNN-based methods for graph coloring on different random graph distributions and the COLOR benchmark. Performance is measured as the number of conflicting edges, with lower indicating better. For each distribution, we report the average over five instances. Our approach outperforms existing methods on most instances and generalizes better to larger and denser graphs. Here $\mathcal{V}$= Nodes, $\mathcal{E}$= Edges, $d$= Average density, $\chi$= Chromatic number.

generate a coloring of the graph with $k$ colors, where $k$ is the chromatic number of the given graph, and report the number of conflicting edges in the generated solution.

**Baselines.** We compare against existing baselines for neural solvers for graph coloring that are trained to generalize to novel graph instances (*GNN-GCP* [41]). We also include a comparison with canonical Graph Neural Networks (*GCN* [39] and *GAT* [62]), and RL guided by Neural Algorithmic Reasoners (*XLVIN* [16]).

**Quantitative Results.** We compare our approach with GNN-based methods for graph coloring. In Table 8, we report the performance on random graphs generated following different distributions and the average performance on the COLOR benchmark. We show that our approach is able to generalize to larger graphs and different distributions better than existing methods. The detailed performance on the COLOR benchmark can be found in Appendix B, where our approach significantly outperforms existing methods. Notably, while GNN-based methods show increasingly worse performance on larger graphs, our method maintains a good performance across scales.

**Qualitative Results.** We present in Figure 7 a graph instance with a valid coloring solution. We visualize the energy map and show that low energy is assigned to all the edges that compose the whole graph. We also show an incorrect solution with two conflicting edges. The energy of the two conflicting edges is higher than the non-conflicting ones.

**Performance with Increased Computation.** In Appendix B we report that increasing the number of particles from 8 to 1024 decreases the average number of conflicting edges from 15.0 to 8.0.

**Ablation Study.** We include in Appendix C ablations on the sampling procedure, which yields 8.0 conflicting edges on average with PEM compared to 12.3 edges with UHMC. Additionally, we ablate the training losses, where we obtain an average of 15.0 conflicting edges with diffusion loss only and 9.0 when using contrastive loss only.

## 4.4 Crosswords

**Setup.** In this section, we report the results on crosswords puzzle solving. Crosswords are word puzzles where letters are arranged in a grid, with words intersecting both horizontally and vertically. Each word is associated with a clue that provides a definition, context or hint for the answer. The goal is to fill the grid so that all words satisfy both the clues and the grid constraints. For our approach, we

**Figure 8: Optimized Samples Across Timesteps and Particles.** We show samples $y^{(t)}$ generated on the crossword puzzle with PEM ($P{=}8$) at timestep $t$ for different particles $P_i$, where $i$ indicates the particle number. The given crossword has five horizontal clues and five vertical clues. At the end of the optimization process, the PEM is able to generate valid solutions to the puzzle. Cells in red indicate incorrect letters.

train a model to generate a valid word given precomputed embeddings of the corresponding hint. To solve a complete crossword, we compose horizontally and vertically the model to form the given grid. We evaluate on the Crosswords Mini Benchmark introduced in [71].

**Baselines.** We compare against different inference algorithms for Large Language Models, including: Standard Input-Output (IO), Chain of Thought (CoT) [65] and Tree of Thought (ToT) [71].

**Quantitative Results.** In Table 9 we compare our approach with various LLM inference methods. Our approach significantly outperforms both the IO and CoT baselines. Furthermore, it achieves performance competitive with ToT. While ToT attains a slightly

| Model | Letter Success Rate | Word Success Rate |
|---|---|---|
| IO | 38.7 | 14.0 |
| CoT | 40.6 | 15.6 |
| ToT | 78.0 | **60.0** |
| EBM (Ours) | **80.4** | 50.5 |

Table 9: **Crosswords Evaluation.** We compare against different strategies for LLM inference. We report the average over 20 crosswords. We used $P{=}1024$ for sampling with PEM.

higher average word success rate (60.0% vs. 50.5%), our method achieves a higher overall grid completion rate (80.4% vs. 78.0%).

**Qualitative Results.** Figure 8 shows the particles generated for a crossword across timesteps. It can be seen that, during the optimization process, different particles explore different solutions to the puzzle. In the end, the optimization algorithm successfully finds a valid solution to the crossword.

## 5  Limitations and Conclusion

**Limitations.** A limitation of our method is that it assumes a starting Gaussian distribution and models optimization as a sequence of Gaussian increments. Future work could explore non-Gaussian objectives and initializations that enable recurrent improvement of initial solutions. Another limitation is that, while our method excels on N-Queens and 3-SAT, there is still room for improvement in achieving optimal solutions for graph coloring. Further research could investigate alternative training strategies for EBMs to produce more accurate energy landscapes

**Conclusion.** In this work, we propose a novel approach to solving reasoning problems using EBMs. By formulating the problem as an energy minimization, we decompose them into smaller subproblems, each with its own energy landscape. We then combine the objective function of each landscape to construct energy landscapes for more complex instances. To train EBMs, we have proposed a combination of diffusion and contrastive loss, which yielded superior performance. We also introduced PEM, a parallel optimization method with an adaptable computation budget for sampling from the resulting energy functions. Our approach outperforms existing methods on several reasoning tasks, including N-queens, 3-SAT, graph coloring, and crossword puzzles, pointing to the promise of this approach.

## Acknowledgments

We thank Siba Smarak Panigrahi for his feedback during the preparation of this manuscript.

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

# Appendix Overview

The Appendix is organized as follows: Section A describes the experimental setting, including hyperparameters and training details, Section B presents additional quantitative and qualitative results on the problems considered in the paper, and Section C includes ablation studies on the sampling procedure and training losses.

# A  Experimental Settings

## A.1  N-Queens Problem

**Setup.**  We used a single instance solution of 8-Queens for both training and validation. For all approaches, the best model was selected based on the validation performance.

**Compositional Approach.**  In our approach, we train a model using the 8 rows from the instance. We use each of the rows as targets to be generated by the model. As negative samples, we used a row without queens and a row with two queens (see Figure 9). In other words, we train a model to generate a vector of length 8 with exclusively one 1 and seven 0s. To compose a 8-queens solver, we simultaneously add the energy of all rows, columns and diagonals using the same model. For the diagonals, we pad the rows with zeros to the right. Notice that this composition, as it is, assumes as a constraint that a queen has to be placed in each diagonal, however, this constraint does not exist in the original problem. Alternatively, we could have trained two separate models: (1) one for the rows and columns, where the constraint is to always have one queen placed, and (2) one for the diagonals, where the constraint is to have either zero or one queen placed. Empirically, we observed that the single model approach led to better heatmaps than training two separate models for rows/columns and diagonals.

**Training.**  As a model, we used a 3-layer MLP, with each layer having: layer normalization and 3 linear layers of dimensions 128, 256, 128, followed by a ReLU activation. We added skip connections for each layer. The model was trained with a learning rate of $1e^{-4}$ with AdamW optimizer for 20000 epochs with a batch size of 2048. For the contrastive loss, we used a weight of 0.5. For scheduled noise we used a linear schedule with $T = 100$ timesteps. With a single Nvidia A10 GPU with 24GB of memory, the model was trained in approximately 5 hours.

**Baselines.**  For all baselines we used the default hyperparameters proposed in each work for MIS solving on SATLIB unless stated otherwise. To encode the N-queens problem as a MIS problem for each position of the chessboard we created a node, and then added an edge with each of the other nodes in the same row, column and diagonal. For LWD we included self-loops for each node. For GFlowNets, we trained a model with 8 layers for 1500 epochs with batch size of 128. For DIFUSCO we trained a model with 8 layers for 5000 epochs. For FastT2T we trained a model with 8 layers for 20000 epochs. In Table 10 we report the wall clock time required to sample 25 solutions of the N-queens problem for all baselines considered. Results are obtained using a single Nvidia A10 GPU with 24GB of memory

## A.2  3-SAT Problem

**Setup.**  We generated 4000 random satisfiable 3-SAT instances for training and 1000 for validation, using the `cnfgen` Python package. For each instance, we randomly sample the number of variables within $[10, 20]$ and assign a number of clauses equal to $4.258 \times n$, where $n$ is the number of variables. This is, we generate samples to be in phase transition , which technically makes it more difficult instances to solve [57]. For evaluation, we used instances from SATLIB, which are also generated to be in phase transition. We used 100 satisfiable instances with 20 variables and 91 clauses, and 100 satisfiable instances with 50 variables and 218 clauses. For all approaches considered, the best model was selected based on validation performance.

**Compositional Approach.**  As a base model, we train a model to generate an assignment that satisfies a single clause. For instance, given a clause $(a \lor b \lor \neg c)$, two example valid assignments are $(1, 0, 1)$ or $(0, 0, 0)$, where $1 = True$ and $0 = False$. Notice that for each clause with three literals, we have seven valid assignments and one invalid assignment. For each clause, as negative sample we use the invalid assignment, which is given by the negation of the clause. For instance, for the previous clause $(0, 0, 1)$ is not a satisfiable assignment. To obtain the clauses for training we split each of the

| Model | Time (s) |
|---|---|
| LwD | $1.16 \pm 0.04$ |
| GFlowNets | $1.12 \pm 0.08$ |
| DIFUSCO | $33.46 \pm 1.36$ |
| FastT2T ($T_S=1, T_G=1$) | $1.82 \pm 0.18$ |
| FastT2T ($T_S=1, T_G=1$, GS) | $1.91 \pm 0.19$ |
| FastT2T ($T_S=5, T_G=5$) | $3.37 \pm 1.12$ |
| FastT2T ($T_S=5, T_G=5$, GS) | $7.78 \pm 0.60$ |
| EBM (P=1024) (Ours) | $84.99 \pm 1.61$ |

Table 10: **8-Queens Problem Inference Time.** We report the wall clock time required to generate 25 solutions of the problem. Values are averaged over five different runs.

| Model | Correct Instances | Unique Solutions |
|---|---|---|
| DeepT | 1 | 1 |
| EBM (P=128) (Ours) | 100 | 37 |

Table 11: **8-Queens Problem Evaluation.** We compare the number of unique solutions generated against the DeepT approach. While DeepT is a deterministic approach, our method generates 37 different solutions to the problem out of 100 correct instances.

| Num. Particles | Energy |
|---|---|
| 2 | $136.62 \pm 2.26$ |
| 4 | $136.29 \pm 1.98$ |
| 8 | $133.93 \pm 1.67$ |
| 16 | $133.82 \pm 2.05$ |
| 32 | $132.47 \pm 1.16$ |
| 64 | $132.36 \pm 0.99$ |
| 128 | $132.00 \pm 1.03$ |
| 256 | $131.84 \pm 0.97$ |

Table 12: **Sampled Energy vs Number of Particles.** A larger number of particles enables sampling lower energy values in the N-Queens problem. Values are averaged over five runs.

| Solution | Energy |
|---|---|
| Random | $151.43 \pm 3.42$ |
| One invalid queen | $137.33 \pm 1.86$ |
| Correct | $130.78 \pm 0.37$ |

Table 13: **Solution Type vs Energy.** We show the comparison of energy values in the 8-Queens problem for three solution types: incorrect solutions with 8 randomly placed queens, nearly-correct solutions with one misplaced queen, and correct solutions. Correct solutions have, on average, the lowest energy values compared to the other solution types. Values are averaged over five runs.

4000 instances into separate clauses. To compose the models, we add the energy of all clauses that make an instance.

**Training.** We trained a 3-layer MLP with skip connections and each layer having layer normalization, 3 linear layers of dimensions 128, 256, 128, followed by a ReLU activation. As input, we use the concatenation of the generated sample (dimension 3) and the clause sign (dimension 3). We trained using the AdamW optimizer with a learning rate of $1e^{-4}$ for 20000 epochs and batch size 1024. We used 0.5 for the contrastive loss weight. For finetuning, we trained for an additional 10000 epochs with a learning rate of $1e^{-4}$ and batch size 1024. For scheduled noise, we used a linear schedule with $T = 100$ timesteps. With a single Nvidia A10 GPU with 24GB of memory, the model was trained in approximately 12 hours.

**Baselines.** For all baselines, we used the default hyperparameters proposed in each work for MIS solving on SATLIB unless stated otherwise. For all the combinatorial optimization models, we modified the architecture to include the clause sign as input. For GCN, we trained a model with 12 layers for 100 epochs with batch size 256. For DIFUSCO, we trained a model for 250 epochs with batch size 256. For FastT2T, we trained a model for 500 epochs with batch size 256. With neural SAT solvers, we used the hyperparameters specified in the corresponding works except for the following: for both NeuroSAT and NSNet, we trained for 300 epochs with batch size 128.

### A.3 Graph Coloring Problem

**Baselines.** We extend the baselines presented in the main paper to include additional comparisons with state-of-the-art reasoning Large Language Models (LLMs). In particular, we consider Gemini 2.5 Pro [14], DeepSeek R1 [31], and Qwen 235B-A22B-2507 [68].

**Setup.** We generated 1000 random graphs following the approach from [41]. The graphs have number of nodes within $[20, 40]$, density within $[0.01, 0.5]$, and chromatic number $\chi$ within $[3, 8]$. We then make a 90-10 split for training and validation. For evaluation, we generated ten random instances from each of the distributions: Erdos-Renyi, Holme-Kim, and random regular expander graphs, with five instances with nodes within $[20, 40]$ and five instances with nodes within $[80, 100]$. Additionally, we generated five Paley graphs with prime numbers between 19 and 37, and complete graphs from 8 to 12 nodes.

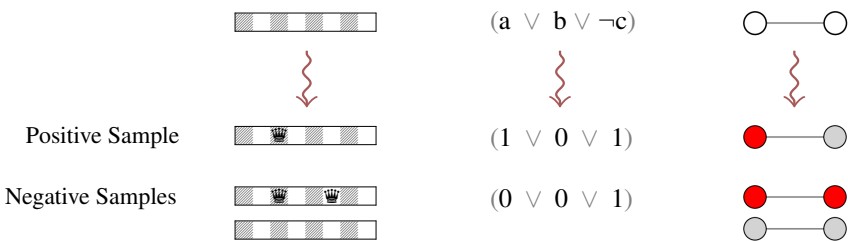

Figure 9: **Negative Samples.** (Left) In the N-Queens problem, the model is trained to generate rows containing exactly one queen. Negative samples consist of invalid rows with no queens or multiple queens. (Middle) For the 3-SAT problem, the model learns to produce valid assignments for a single clause. The negative sample used is obtained directly by negating the clause sign. (Right) In the graph coloring problem, the model is trained to assign different colors to the nodes of an edge. Edges with the same color for both nodes are used as negative samples.

**Compositional Approach.** We trained a base model to generate a valid coloring for a single edge. We define a fixed set of colors, in this case $k=14$ colors. The model generates a one-hot vector encoding the left and right colors of the edge (in total, an output of dimension 28). As negative samples, we used edges with the same color for both nodes. Additionally, to enforce the generation of valid one-hot vectors, we also used as negatives vectors with random perturbations in a random position. To compose the model for the whole graph, we add the energy of all edges that compose the graph.

**Training.** We trained a 4-layer MLP with skip connections and each layer having layer normalization, 3 linear layers of dimensions 128, 256, 128, followed by a ReLU activation. We trained using the AdamW optimizer with a learning rate of $1e^{-4}$ for 50000 epochs and batch size 1024. We used 0.5 for the contrastive loss weight. For scheduled noise, we used a linear schedule with $T = 100$ timesteps. With a single Nvidia A10 GPU with 24GB of memory, the model was trained in approximately 3 hours.

**Baselines.** For both GCN and GAT, we trained a model with 8 layers with hidden dimension 128, dropout 0.1 with AdamW for 1000 epochs and batch size 512. We train the models using cross-entropy loss to predict the color of each node out of 14 colors. To train GNN-GCP we follow the same setting as in the original work and stop the training when the model achieves 82% accuracy and 0.35 binary cross-entropy loss on the training set averaged over 128 batches containing 16 instances. For XLVIN, we train the XLVIN-CP variant using CartPole-style synthetic graphs and use the same hyperparameteras reported in the original work.

### A.4 Crosswords

**Setup.** We evaluate approaches based on their ability to solve 5x5 crosswords with 10 words each. For evaluation, we use the 20 crosswords from the Mini Crosswords dataset from [71]. To train our approach, we sample 32.7k and 6.8k five-letter words from the Crosswords QA dataset [63] for training and validation, respectively. Each entry in the dataset consists of a hint and a five-letter word. To generate embeddings of the hints, we use the `text-embedding-3-small` model from OpenAI.

**Compositional Approach.** As a base model, we train a model that, given the embedding of a hint, is able to generate a five-letter word. We later compose the model to solve a 5x5 crossword by combining the energy functions of the five horizontal rows and five vertical columns, where each of the ten words has a separate hint as input.

**Training.** As a model, we use a 3-layer MLP, with each layer having layer normalization and three linear layers of dimensions 1024, 1024, and 1024, followed by a ReLU activation. We add skip connections for each layer. The model receives embeddings of dimension 1536 as input. The model is trained with a learning rate of $1 \times 10^{-4}$ using the AdamW optimizer for 20,000 epochs with a batch size of 2048. We did not use contrastive loss for training the model. For scheduled noise, we use a linear schedule with $T=100$ timesteps.

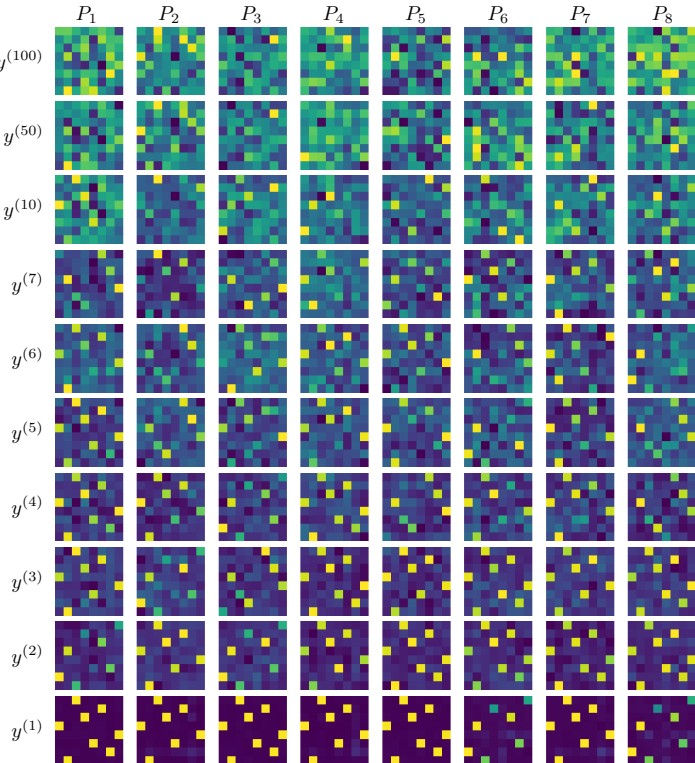

Figure 10: **Optimized Samples Across Timesteps and Particles.** We show samples $y^{(t)}$ generated on the 8-queens problem with PEM ($P{=}8$) at timestep $t$ for different particles $P_i$, where $i$ indicates the particle number. In the figure, yellow squares represent queens placed in the chessboard. PEM is able to generate a valid instance of the 8-queen problem ($y^{(1)}$ with $P_1, P_3$ and $P_5$). A first valid solution appears at $y^{(5)}$ with $P_1$.

# B    Additional Results

## B.1    N-queens Problem

**Quantitative Results.**  In Table 11 we provide a quantitative comparison where we compare with Deep Thinking (DeepT [46]). While we were able to successfully solve the 8-queens problem using both approaches, the DeepT approach is purely deterministic, producing the same solution each time. In the case of N-queens, where there are multiple solutions, and no input is provided, this is a limitation. Using our approach, we are able to generate 37 unique solutions out of 100 sampled solutions.

**Qualitative Results.**  In Figure 10 we show the samples generated with PEM at different timesteps and particles. It can be seen that at the end of the sampling process, the procedure reaches a valid solution for the 8-queens problem. During the optimization process, it can be seen how different particles partially approximate different solutions until they converge to the same solution.

## B.2    3-SAT Problem

**Qualitative Results.**  In Figure 11 we consider an instance of the 3-SAT problem with four clauses and three variables. We compare two variable assignments: a correct assignment satisfies all clauses, and an incorrect one that satisfies only three. The figure shows the energy computed by the model for each clause individually. In the correct solution, all clauses are assigned low energy. In contrast, in the incorrect solution, the unsatisfied clause has comparatively higher energy than the others. This highlights how the energy function effectively reflects clause satisfaction. As a consequence, when the energy of all clauses is composed, a higher energy is assigned to the incorrect solution.

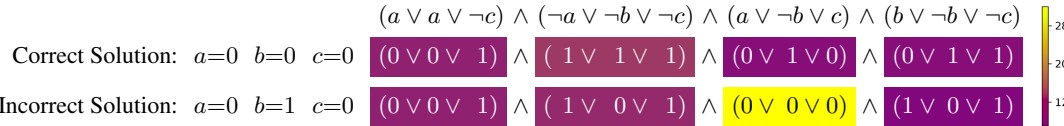

Figure 11: **Qualitative Visualization of Energy Maps.** Example of a 3-SAT instance with four clauses and three variables (top), along with a correct solution (middle) and an incorrect solution (bottom), having 1=True and 0=False. We show the energy of each clause individually. A higher energy is assigned to the clause that evaluates to false (unsatisfied clause), while clauses that evaluate to true (satisfied clauses) are assigned lower energy.

| | Conflicting Edges ↓ | |
|---|---|---|
| Model | Erdos-Renyi Small | Erdos-Renyi Large |
| Gemini 2.5 Pro | $20.00 \pm 14.22$ | $142.66 \pm 6.11$ |
| Deepseek R1 | $4.00 \pm 3.46$ | $78.00 \pm 36.38$ |
| Qwen 235B-A22B-2507 | $16.00 \pm 13.06$ | $102.66 \pm 26.10$ |
| PEM (P=128) (Ours) | $3.15 \pm 2.00$ | $45.81 \pm 11.88$ |

Table 15: **Comparison of Conflicting Edges across Models.** Lower values indicate fewer conflicts. PEM (ours) achieves the lowest number of conflicting edges across both settings. Values are averaged over five graphs.

## B.3 Graph Coloring Problem

**Baslines**

**Quantitative Results.** In Table 16 we provide an evaluation of our approach on the COLOR benchmark. We compare against existing methods for graph coloring methods and canonical GNNs. We can see that methods based on GNNs generalize worse with increasingly larger graphs. On the larger graph considered, our approach generates a solution with 69 conflicting edges, while GNN-GCP generates a solution with 667 conflicting edges, and GCN and GAT generate solutions with 1625 and 1454 conflicting edges, respectively. In Table 15 we compare our approach with state of the art reasoning Large Language Models on graphs following the Erdos-Renyi distribution. While some models are able to solve complete instances in-context (e.g. DeepSeek achieves an average of 4.0 conflicting edges on small Erdos-Renyi graphs), their performance deteriorates as graph

| Num. Particles | Conflicting Edges ↓ |
|---|---|
| 8 | $15.0 \pm 2.64$ |
| 64 | $14.3 \pm 3.51$ |
| 128 | $10.3 \pm 2.51$ |
| 1024 | $8.0 \pm 2.64$ |

Table 14: **Number of Particles vs Conflicting Edges.** We sampled three solutions for a given graph instance. Increasing the number of particles with PEM leads on average to more optimal solutions.

size increases. For larger graphs, these models fail to find effective in-context solutions, with the best achieving an average of 78.0 conflicting edges compared to 45.81 with our approach.

**Qualitative Results.** In Figure 12 we show the samples generated with PEM at different timesteps and particles. We can see that, at timestep $t = 30$, particle $P_4$ generates the first valid solution and that, at timestep $t = 10$, all particles have already converged to a valid solution. In Figure 13 we show an example energy landscape resulting from the composition of two edges, where the optimal solution corresponds to the minimum of the function. Additionally, in Figure 14 we show the evolution of the landscape over different timesteps.

**Performance with Increased Computation.** We assess the effect of increased computation in Tables 14 and 17. We find that a larger number of particles in sampling slightly improves performance on the graph coloring task. On average, the number of conflicting edges in the generated solution is lower with a larger number of particles, meaning that the solution is closer to the optimal solution.

**Performance with Increased Timesteps.** In Table 18 we evaluate the effect of the timesteps hyperparameter in the graph coloring performance using the Holme Kim distribution. We observe that increasing the number of timesteps from 20 to 1000 leads to an average reduction in the number of conflicting edges, from 6.89 to 6.51 in small instances, and from 44.27 to 42.71 in larger instances.

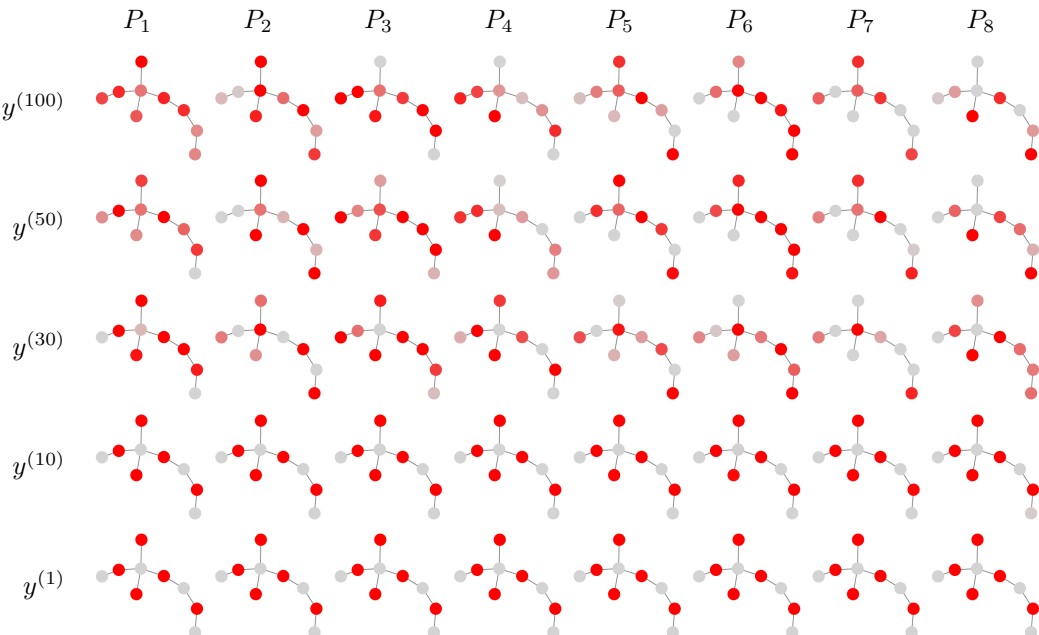

Figure 12: **Optimized Samples Across Timesteps and Particles.** We show samples $y^{(t)}$ generated on the graph coloring problem with PEM ($P = 8$) at timestep $t$ for different particles $P_i$, where $i$ indicates the particle number. The graph instance has eight edges, nine nodes and chromatic number $\chi=2$. PEM is able to generate a valid coloring for the graph (red and gray in the figure).

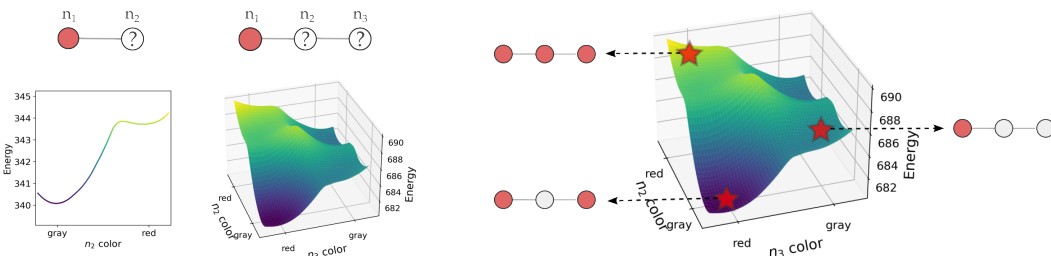

Figure 13: **Composition of Energy Landscapes.** (Left) Energy landscape for different values of $n_2$. from a simple graph coloring problem with one edge and fixed color red for one node. The plot shows that the energy assigned to gray color is the lowest. By composing two energy landscapes, we can create a new function corresponding to a larger problem with two edges. (Right) The energy landscape resulting from composing the energy landscapes of two edges with one node fixed to color red. The plot shows the energy for combinations of colors for nodes $n_2$ and $n_3$. The assignment $n_2$=gray and $n_3$=red yields the lowest energy, indicating that this is the optimal solution.

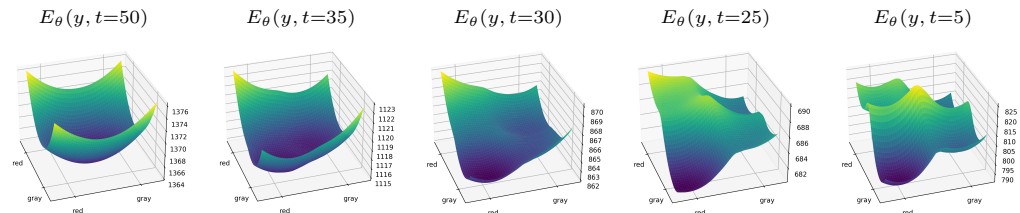

Figure 14: **Energy Landscapes Across Timesteps.** Evolution of energy landscapes over time for a graph coloring problem with two edges (shown in Figure 13). The landscapes transform a Gaussian distribution into gradually the target distribution. Eventually, the optimal solution becomes the global minimum of the function.

| Graph | $\mathcal{V}$ | $\mathcal{E}$ | $d$ | $\chi$ | GCN | GAT | GNN-GCP | EBM (Ours) (P=128) |
|---|---|---|---|---|---|---|---|---|
| myciel3 | 11 | 20 | 0.36 | 4 | 17 | 10 | 6 | 2 |
| myciel4 | 23 | 71 | 0.28 | 5 | 71 | 31 | 10 | 9 |
| queen5_5 | 25 | 160 | 0.53 | 5 | 160 | 160 | 96 | 23 |
| queen6_6 | 36 | 290 | 0.46 | 7 | 290 | 290 | 116 | 37 |
| myciel5 | 47 | 236 | 0.22 | 6 | 171 | 97 | 42 | 26 |
| queen7_7 | 49 | 476 | 0.40 | 7 | 476 | 476 | 216 | 64 |
| queen8_8 | 64 | 728 | 0.36 | 9 | 728 | 728 | 272 | 73 |
| 1-Insertions_4 | 67 | 232 | 0.10 | 5 | 220 | 100 | 42 | 21 |
| huck | 74 | 301 | 0.11 | 11 | 216 | 234 | 172 | 20 |
| jean | 77 | 254 | 0.09 | 10 | 208 | 146 | 206 | 14 |
| david | 87 | 406 | 0.11 | 11 | 333 | 352 | 156 | 26 |
| mug88_1 | 88 | 146 | 0.04 | 4 | 117 | 127 | 98 | 17 |
| myciel6 | 95 | 755 | 0.17 | 7 | 755 | 755 | 100 | 98 |
| queen8_12 | 96 | 1368 | 0.30 | 12 | 1368 | 1368 | 408 | 96 |
| games120 | 120 | 638 | 0.09 | 9 | 574 | 438 | 418 | 52 |
| anna | 138 | 493 | 0.05 | 11 | 260 | 392 | 110 | 34 |
| 2-Insertions_4 | 149 | 541 | 0.05 | 5 | 270 | 304 | 198 | 64 |
| homer | 556 | 1629 | 0.01 | 13 | 1625 | 1454 | 667 | 69 |

Table 16: **Graph Coloring Evaluation.** We compare the performance against canonical GNNs and GNN-based methods for graph coloring on the COLOR benchmark. For each graph, we report the number of conflicting edges in the coloring solution, where lower is better. Our methods outperform existing methods on almost all the instances and shows better generalization to larger graphs.

| | Conflicting Edges ↓ | |
|---|---|---|
| Num. Particles | Holme Kim Small | Holme Kim Large |
| 128 | $10.60 \pm 2.70$ | $59.00 \pm 5.24$ |
| 256 | $8.38 \pm 3.28$ | $56.89 \pm 12.75$ |
| 512 | $9.68 \pm 3.13$ | $54.68 \pm 8.10$ |
| 1024 | $7.04 \pm 3.64$ | $57.04 \pm 3.27$ |
| 2048 | $7.79 \pm 4.20$ | $55.57 \pm 4.58$ |

Table 17: **Number of Particles vs Conflicting Edges.** We report solutions averaged over five instances of each distribution. Increasing the number of particles with PEM on average decreases the number of conflicting edges in Holme Kim graph distributions.

| | Conflicting Edges ↓ | |
|---|---|---|
| Num. Timesteps | Holme Kim Small | Holme Kim Large |
| 20 | $6.98 \pm 2.19$ | $44.27 \pm 7.29$ |
| 50 | $7.01 \pm 2.68$ | $54.40 \pm 8.61$ |
| 100 | $6.71 \pm 2.60$ | $52.97 \pm 8.07$ |
| 200 | $6.00 \pm 3.83$ | $50.31 \pm 5.76$ |
| 500 | $6.50 \pm 2.58$ | $49.63 \pm 3.19$ |
| 1000 | $6.51 \pm 2.35$ | $42.71 \pm 3.93$ |

Table 18: **Number of Timesteps vs Conflicting Edges.** Training with a higher number of timesteps decreases on average the number of conflicting edges in Holme Kim graph distributions. We report solutions averaged over five instances of each distribution. In all cases we sample using PEM ($P=1024$).

| | Similar Distribution | | Larger Distribution | |
|---|---|---|---|---|
| **Sampler** | **Correct Instances ↑** | **Satisfied Clauses ↑** | **Correct Instances ↑** | **Satisfied Clauses ↑** |
| Reverse Diffusion | 1 | 0.9521 | 0 | 0.9519 |
| ULA | 0 | 0.9524 | 0 | 0.9516 |
| MALA | 0 | 0.9519 | 0 | 0.9535 |
| UHMC | 1 | 0.9502 | 0 | 0.9537 |
| HMC | 1 | 0.9553 | 0 | 0.9533 |
| EBM ($P = 1024$) | 91 | 0.9985 | 43 | 0.9963 |

Table 19: **Sampler Ablation.** Ablations proposed for samplers on the 3-SAT task. PEM significantly outperforms other samplers on the 3-SAT problem for both similar and larger distributions.

| Diffusion Loss | Contrastive Loss | Correct Instances ↑ | Satisfied Clauses ↑ |
|---|---|---|---|
| No | Yes | 0 | $0.9331 \pm 0.0258$ |
| Yes | No | 11 | $0.9742 \pm 0.0157$ |
| Yes | Yes | 57 | $0.9951 \pm 0.0068$ |

Table 20: **Loss Ablation.** Ablations proposed for the loss function on the performance on the 3-SAT problem. A combination of both a diffusion loss to train the EBM and a contrastive loss to shape the landscape leads to the best results. In all cases we sampled using PEM ($P{=}1024$).

| **Sampler** | **Conflicting Edges ↓** |
|---|---|
| Reverse Diffusion | $19.6 \pm 3.51$ |
| ULA | $28.3 \pm 4.93$ |
| MALA | $17.0 \pm 2.64$ |
| UHMC | $12.3 \pm 2.51$ |
| HMC | $14.6 \pm 0.57$ |
| PEM ($P = 1024$) | $8.0 \pm 2.64$ |

Table 21: **Sampler Ablation.** Ablations proposed for different samplers on the graph coloring task. We sample three solutions for a given graph instance. On average PEM finds solution with a lower number of conflicting edges.

| Diffusion Loss | Contrastive Loss | Conflicting Edges ↓ |
|---|---|---|
| No | Yes | $9.0 \pm 2.00$ |
| Yes | No | $15.0 \pm 4.00$ |
| Yes | Yes | $8.0 \pm 2.64$ |

Table 22: **Loss Ablation.** Ablations proposed for the loss function on the performance on the graph coloring task. We sample three solutions for a given graph instance. A combination of both diffusion and contrastive loss leads to the best results. In all cases we sampled using PEM ($P{=}1024$).

## C  Ablation Study

**3-SAT Problem.** We ablate the sampling procedure in Table 19. We compare the performance of our method with different samplers. In the similar distribution setting, our method successfully finds satisfiable solutions in 91 out of 100 instances, whereas baseline samples find at most one. In the larger distribution setting, our method finds 43 satisfiable solutions out of 100, while baseline samples do not succeed. We also ablate the training losses in Table 20. A model trained with both diffusion and contrastive loss solves 57 out of 100 instances, compared to 11 with diffusion loss only and 0 with contrastive loss only.

**Graph Coloring Problem.** We propose ablations for both training and sampling of EBMs on the graph coloring task. In Table 21, we show that on average our sampling procedure produces on average more optimal solutions. In Table 22 we ablate the diffusion loss and contrastive loss used to train the model. The combination of both losses leads to the best performance.

