# OpenReview forum: "Generalizable Reasoning through Compositional Energy Minimization"
_NeurIPS.cc/2025/Conference — NeurIPS 2025 spotlight_

### Official Review · Reviewer_7Yz8 · 2025-07-01

**Clarity:** 3
**Significance:** 2
**Originality:** 3
**Rating:** 4
**Confidence:** 4

**Summary:**

This paper proposes Parallel Energy Minimization (PEM) a particle-based optimization strategy for sampling at inference time. The key hypothesis is that valid (correct) solutions would be assigned lower energy and invalid/incorrect solutions would have high energy. The method essentially conducts search by decomposing the problem, i.e., the energy function into more tractable sub-problems that are optimized in parallel. The authors test the effectiveness of their method on three algorithmic/compositional tasks: N-Queens, 3-SAT and graph coloring with the common theme of given a question generating an answer (final state configuration) that satisfies the problem constraints and find that PEM outperforms other sampling/search baselines.

**Questions:**

- Is it a reasonable/realistic assumption to know the test time complexity ahead of time? If not how should the number of particles be set for new tasks?
- Lots of reasoning tasks have unique final answer but several correct ways  or chains to achieve them. For the tasks studied in the paper the correct configuration/state may also may not be unique -- do these have the same energy -- how does energy map to correctness?
- Seems like the method would only work for algorithmic formalization. Would the sampling also be useful if the constraints and actions (large k in graph coloring, closest which makes the problem easier and not harder) are in text or with a large action space? Results on tasks such as BlocksWorld, crossword, and Game-24 would improve the strength of this work.

**Ethical Concerns:**

["NO or VERY MINOR ethics concerns only"]

**Final Justification:**

Increased the score since I find the results from the crosswords experiment convincing

**Limitations:**

Yes.

**Paper Formatting Concerns:**

It appears that the hyperlinks in the citations are not working as intended.

**Quality:**

3

**Strengths And Weaknesses:**

**Strengths:**
- The paper is well-written, easy to follow, and addresses the important topic of sampling strategies dedicated to "reasoning" tasks that require planning and search to some extent. The method itself seems intuitive.
-  The method achieves impressive results especially on N-queens and 3-SAT tasks along with ablations that back up the effectiveness of their design.

**Weaknesses:**
- The reasoning emphasis in the introduction, abstract, and title are misleading and not consistent with the nature of the method or tasks studied. The paper predominately focuses on algorithmic tasks with search and constraint satisfaction components. However, the aforementioned section give the impression that method can be used for textual reasoning tasks (like the ones frontier LLMs excel at) and per my understanding their method would not easily generalize to such tasks.
- Complexity of the tasks studied is not be high enough: most of the tasks have a binary action space/constraints. See question below. Establishing that their method works on more planning or search-based benchmarks with a larger action space would improve the strength of the submission.

---

> ### Author Rebuttal · Authors · 2025-07-31
>
> We thank the reviewer for appreciating the clarity of the paper and results reported in the paper. We believe the reviewer has made significant points that will help us make a stronger submission. We address these concerns in detail below:
>
> ### **Q1) Number of particles for new tasks**
>
> We agree that the number of particles is a hyperparameter that is not straightforward to set for new tasks. One potential option for setting the number of particles is to increase the number adaptively until convergence is achieved. For instance, one can use the energy of the sampled solution as a measure of convergence, and increase the number of particles until the energy does not change significantly. To provide an example, the following table shows the average energy of the sampled solutions for the 8-Queens problem, increasing from 2 particles to 256. It can be seen that after 32 particles, the difference in energy compared to the previous value becomes, on average, less than 0.1, which can be used as a stopping criterion.
> | Num. Particles | Energy       	|
> | -------------- | ----------------- |
> | 2          	| $136.62 \pm 2.26$ |
> | 4          	| $136.29 \pm 1.98$ |
> | 8          	| $133.93 \pm 1.67$ |
> | 16         	| $133.82 \pm 2.05$ |
> | 32         	| $132.47 \pm 1.16$ |
> | 64         	| $132.36 \pm 0.99$ |
> | 128        	| $132.00 \pm 1.03$ |
> | 256        	| $131.84 \pm 0.97$ |
>
> ### **Q2) Energy and correctness relationship**
>
> This is a significant point in the work. The presented method is based on the assumption that, if one of the models that is being composed is not satisfied or solved, the energy of the solution will be higher, while if the model is satisfied, the energy will be lower. As a consequence of this, a correct solution, i.e., a solution where all the subproblems are satisfied, will have the lowest energy. If at least one of the subproblems is not satisfied, the corresponding energy will be higher and, as a consequence, the composed energy will also be higher. To provide quantitative results on the energy of the solutions, we use the 8-Queens problem as an example, with the table below showing the energy of the solutions for the 8-Queens problem at timestep $T = 0$. We compare the following solutions:
> 1. The 92 correct solutions to the 8-Queens problem.
> 2. 92 solutions with 8 queens where at least one queen is not placed in a valid position.
> 3. 92 random solutions where 8 queens are placed randomly.
>
> Each of the entries reports the energy averaged over the 92 instances:
>
> | Solution Type | Energy        	|
> | -----------------|-----------------|
> | Random   | $151.43 \pm 3.42$ |
> | One invalid queen | $137.33 \pm 1.86$ |
> | Correct | $130.78 \pm 0.37$ |
>
> From this, we can observe that all the correct solutions of the 8-Queens problem share a similar energy, and that, moreover, this is the minimum energy if we compare it to other suboptimal solutions.
>
>
> ### **Q3) Additional benchmark**
>
> Regarding the search space of the problems studied, notice that for some of the graph coloring problems, the action space is considerably large. For instance, for graph "homer" from the COLOR benchmark, with 1629 edges and k=13, the resulting search space is of $1629 \times 13$ binary variables.
>
> We agree that the method can also be applied to additional tasks to improve the strength of the work. To this end, here we provide additional evaluation on the Crosswords tasks.
>
> - **Crosswords:**
> We evaluate using the Mini Crosswords dataset from [3]. This dataset is a collection of 5x5 crossword puzzles. Each crossword puzzle has five clues for the rows and five clues for the columns. To be able to generate the given words, we used the CrosswordsQA datasets from [4]. From this dataset, we extracted only the words that appear in the Mini Crosswords dataset, which resulted in 38113 pairs of clue words. We then trained our EBM to predict a given word given a clue. To encode the clues, we used embeddings from the text-embedding-ada-002 model from OpenAI.
> To solve a 5x5 crossword puzzle, we compose the trained model spatially, forming the five rows and five columns of the crossword puzzle. We then sample from the composed model using PEM.
>
>   We compare our approach against the results from GPT-4 [3], including:
>   - Standard Input-Output (IO)
>   - Chain of Thought (CoT)
>   - Tree of Thoughts (ToT) [3].
>
>   In the table below, we report the success rate at the level of letters and words. We show that, despite not exceeding current results from ToT, our approach shows competitive performance in a complex search space task, with 72% of words correctly generated, compared to 78% from ToT. Additionally, we show that our method significantly outperforms both IO and CoT with 43% of correct words versus 14% and 15% respectively. This result is especially relevant, taking into account that GPT-4 has been trained on a large corpus of text data and contains significantly more semantic knowledge than our model.
>
>   | Model      	| Letter Success Rate | Word Success Rate |
>   | ---------------| -------------------- | ------------------ |
>   | IO                 | 38.7               | 14.0               |
>   | CoT               |  40.6              | 15.6               |
>   | ToT        	| 78.0             	| 60.0           	|
>   | EBM (P=256) (ours) |  72.0            	|  43.0              	|

---

> > ### Comment · Reviewer_7Yz8 · 2025-08-03
> >
> > In light of the new crosswords result, I am happy to increase the score of the paper.

---

> > > ### Author Response · Authors · 2025-08-04
> > >
> > > We are grateful to the reviewer for appreciating our additional results
> > > on the crosswords benchmark.
> > > We would also like to thank the reviewer for their effort and time in reviewing the paper.
> > > We will update the paper with the novel results and comparisons provided.

---

### Official Review · Reviewer_6UFB · 2025-07-02

**Clarity:** 3
**Significance:** 4
**Originality:** 3
**Rating:** 5
**Confidence:** 3

**Summary:**

This work proposes a novel energy-based method for generating solutions to reasoning problems by learning energy landscapes of subproblems and combining them. The method combines diffusion and contrastive loss, and is evaluated against state-of-the-art methods on problems like N-queens, SAT, and graph coloring. All methods are trained and then compared in out-of-distribution setting. Proposed method significantly outperforms existing SoTA methods.

**Questions:**

- What are the limits on performance of increasing the number of particles?
- What is the computational time of your method compared to other compared methods?
- How are the N-queens decomposed into subproblems? Is your method applicable to tasks that are not decomposable?

**Ethical Concerns:**

["NO or VERY MINOR ethics concerns only"]

**Final Justification:**

Authors answered questions, added requested experiments, and argued well for certain design choices of this work. As it was pointed out by other reviewers, measuring the method's performance on a wider problem variety would be beneficial to this work. Additionally, the method is quite computationally intensive, although this is somewhat warranted due to the strong results. Overall, i suggest to accept this paper, but lower my score from 6 to 5 due to the previously mentioned limitations.

**Limitations:**

Yes

**Paper Formatting Concerns:**

No concerns

**Quality:**

4

**Strengths And Weaknesses:**

Strengths
- The paper is well written, with clear formulas and descriptions of the method. The supplement contains sufficient details about implementation and experiments.
- The training method is novel and a major improvement over existing SoTA methods, while allowing for some interpretability through plotting energy values of solutions.

Weaknesses
- Increasing the number of particles seems to always improve performance. Limits of increasing this parameter to improve performance are not explored.
- The computational time of the method is not compared against other methods.
- There is a minor mistake in equation 3 notation, where the expectation should mark that it's over ε ∼ N (0, I), not N(ε,0,I)
- When solving specific problems like N-Queens, it is not entirely clear how the algorithm decomposes them into subproblems.

---

> ### Author Rebuttal · Authors · 2025-07-31
>
> We thank the reviewer for appreciating the novelty of our method and the clarity of our paper. The reviewer has raised interesting points that will help us improve the paper.
> We address the questions below:
>
> ---
>
> ### **Q1) Scaling the number of particles**
>
> We have scaled the results from the graph coloring problem, particularly those from the Holme-Kim graph distribution, up to our hardware and time limits. We report the results in the table below. In this table, we show that increasing the number of particles does lead to an improvement in the number of conflicting edges, from 59.00 on average to 55.57 in the case of the larger instances. Notice, however, that we double the number of particles for each entry of the table to obtain improved results. That is, while there is some potential for further improvement of the results, one needs to take into account the computational cost of increasing the number of particles. We will add this additional information to the appendix of the paper.
>
> | Num. Particles | Conflicting Edges (Holme-Kim [22-34 nodes]) | Conflicting Edges (Holme-Kim [86-100 nodes]) |
> |---------------| -------------------------------------------- | --------------------------------------------- |
> | 128        	| $10.60 \pm 2.70$          | $59.00 \pm 5.24$ |
> | 256        	| $8.38 \pm 3.28$           	| $56.89 \pm 12.75$ |
> | 512        	| $9.68 \pm 3.13$           	| $54.68 \pm 8.10$  |
> | 1024       	| $7.04 \pm 3.64$           	| $57.04 \pm 3.27$  |
> | 2048       	| $7.79 \pm 4.20$           	| $55.57 \pm 4.58$  |
>
> ---
>
> ### **Q2) Computational time**
> To provide a comparison of the computational time, in the table below we report the wall clock time required to solve 25 instances of the 8-queens problem:
>
> | Model      	| Time (s)     	|
> | ------------------| -----------------------|
> | LwD        	| $1.16 \pm 0.04$  |
> | GFlowNets  	| $1.12 \pm 0.08$  |
> | DIFUSCO    	| $33.46 \pm 1.36$  |
> | FastT2T ($T_S = 1, T_G = 1$) | $1.82 \pm 0.18$ |
> | FastT2T ($T_S = 1, T_G = 1$, GS) | $1.91 \pm 0.19$ |
> | FastT2T ($T_S = 5, T_G = 5$) | $3.37 \pm 1.12$ |
> | FastT2T ($T_S = 5, T_G = 5$, GS) | $7.78 \pm 0.60$ |
> | EBM (P=1024) (ours) | $84.99 \pm 1.61$ |
>
> It can be observed that, compared to other methods, our approach shows a comparable overhead in terms of computational time. This is expected, taking into account that during inference, we sample from multiple composed models instead of a single monolithic model. While we show that this approach leads to better generalization, there is a trade-off in terms of computational time. Note that, since we use a diffusion objective for training, multiple methods that exist in the literature for speeding up diffusion models can be potentially applied to our method too. We will include this additional information in the appendix of the paper.
>
> ---
>
> ### **Q3) N-queens decomposition**
>
> The N-queens problem can be decomposed into the constraints that a solution must satisfy, that is, no queen can be placed in the same row, column or diagonal as another queen. This can be formulated as the following constraints: each row should have exactly one queen, each column should have exactly one queen, and each diagonal should have at most one queen. If we train a model that can satisfy these two constraints:
> 1. Place one queen in each row.
> 2. Place at most one queen on a diagonal.
>
> We can then arrange these two models forming a chessboard and sample from the composition to obtain a solution to the N-queens problem. In the end, model 1 is repeated 8 times (for the rows), 8 times (for the columns) and model 2 is repeated 15 times (for the diagonals). Empirically, we found that using model 1 for the 15 diagonals provided decent results too. We think including an additional visualisation of this decomposition in the paper would significantly improve the clarity for the reader.
>
> Regarding the applicability to other tasks, composition is the means by which we achieve generalization in this work. The method presented in this work is potentially also applicable to e.g. composing multiple EBMs, even if they are not components of a decomposed problem. In this work in particular, we focus on decomposable tasks to show that by learning simpler instances, we are able to combine them to solve larger or out-of-distribution instances.

---

> > ### Comment · Reviewer_6UFB · 2025-08-02
> >
> > Thank you to the authors for comprehensively addressing a variety of reviewers' questions and measuring the computational time of the method solving 8-queens. I appreciate the addition of comparisons to reasoning LLMs and benchmarks on the Mini Crosswords dataset. As pointed out by other reviewers, benchmarking on a larger variety of more complex problems would strengthen this work.

---

> > > ### Author Response · Authors · 2025-08-04
> > >
> > > We are grateful to the reviewer for appreciating the additional results and comparison provided.
> > > We would also like to thank the reviewer for their effort and time in reviewing our manuscript.
> > > We will update our paper accordingly to include the suggested results, as well as the
> > > benchmarks and comparisons with reasoning LLMs

---

### Official Review · Reviewer_xJaU · 2025-07-02

**Clarity:** 3
**Significance:** 2
**Originality:** 2
**Rating:** 5
**Confidence:** 4

**Summary:**

This paper discusses a framing for reasoning based on energy minimization. The authors propose parallel energy minimization, a method that composes an energy landscape given a new problem instance.

**Questions:**

See the weaknesses section. I'm excited to engage with the authors and I hope that with some iteration, this paper becomes a strong submission.

**Ethical Concerns:**

["NO or VERY MINOR ethics concerns only"]

**Final Justification:**

The authors addressed my concerns and provided clarity and new results in response to my questions.

**Limitations:**

Yes.

**Paper Formatting Concerns:**

No.

**Quality:**

3

**Strengths And Weaknesses:**

Strengths:
- Interesting approach to reasoning.


Weaknesses:
- Limited benchmark datasets. The modern setting for reasoning is missing from this paper. Recent work on AI reasoning models use long form open ended math, programming, and science questions. I acknowledge the importance of non-LLM work, but the questions remains, can state-of-the-art reasoning LLMs solve the N-Queens problem or the graph coloring problem? Do we still need specialist models for this type of problem?
- Formatting/latex issue. I printed this paper to read and mark up and none of the instances of "fi" print correctly. In fact, even in my PDF viewer application, the find function fails to find any instances of "fi". I'm extremely curious if the authors can respond to this. Is this because of a tex issue you know about? Is it the result of some formatting issue that might be a problem for the conference template?
- Limited baselines. How do existing specialist iterative reasoning models do on these problems. I'm thinking of models like those in [7] for N-Queens and in https://proceedings.mlr.press/v202/bevilacqua23a/bevilacqua23a.pdf and https://proceedings.neurips.cc/paper/2021/file/82e9e7a12665240d13d0b928be28f230-Paper.pdf for graph coloring.

---

> ### Author Rebuttal · Authors · 2025-07-31
>
> We thank the reviewer for finding our approach interesting and for the constructive feedback. We believe that the feedback is useful and will help us improve the paper. We discuss the concerns raised below:
>
> ---
>
> ### **Q1) Comparison with reasoning LLMs**
>
> We agree that LLM reasoning shows impressive results in certain domains. While we also agree on the importance of non-LLM approaches, we want to note that the approach presented in this paper is not necessarily limited to the algorithmic domain and can be adapted to other modalities too (e.g. text, images, etc.).
>
> We also agree that, given the results of LLMs in reasoning, a comparison would improve the paper coverage of current reasoning approaches. To this end, here we discuss the comparison of our method against reasoning LLMs on the N-Queens problem and the graph coloring problem.
>
> - **N-Queens:** We observed that all the LLMs we evaluated (see graph coloring below) have memorized at least one solution instance from 8-queens up to 16-queens. This makes the comparison difficult, as it is hard to know whether the LLMs are actually reasoning or just recalling a memorized solution.
>
> - **Graph coloring:** We evaluated using random Erdos-Renyi graphs, which guarantees that the LLMs do not have memorized solutions. We provide quantitative results below on a smaller graph with 38 nodes, 146 edges and k=3, and a larger graph with 96 nodes, 386 edges and k=3. We observe that some LLMs, such as Qwen-235B, are, in some cases, able to solve the smaller graph by backtracking fully in-context. However, once the graph size increases, the context window becomes a limitation, and the models usually output random guesses or heuristic guesses based on the degree of the nodes. These results indicate that while LLMs can reason over smaller, more tractable instances, solving larger or more complex reasoning problems remains a limitation for them as of today. We will include these additional results in the appendix of the paper.
>
>     | Model                | Conflicting Edges (38 nodes)     | Conflicting Edges (96 nodes)     |
>     |----------------------|----------------------------------|----------------------------------|
>     | Gemini 2.5 Pro       | $20.00 \pm 14.22$                | $142.66 \pm 6.11$                |
>     | Deepseek R1          | $4.00 \pm 3.46$                  | $78.00 \pm 36.38$                |
>     | Qwen 235B-A22B-2507  | $16.00 \pm 13.06$                | $102.66 \pm 26.10$               |
>     | PEM (P=128) (ours)   | $3.15 \pm 2.00$                  | $45.81 \pm 11.88$                |
>
> ---
>
> ### **Q2) Issue with “fi” formatting**
>
> Unfortunately, we were unable to reproduce the issue with the "fi" characters. We printed the paper and used different PDF viewers, and in all cases, the "fi" characters printed correctly. We have used the default formatting packages included in the Neurips template: `\usepackage[utf8]{inputenc}`, `\usepackage[T1]{fontenc}` and `\usepackage{microtype}`. The document was compiled using PDFLaTeX. If the reviewer has any additional information on how to reproduce the issue, we are happy to update the paper to solve this issue.
>
> ---
>
> ### **Q3) Comparison with iterative/neural algorithmic reasoning**
>
> Thank you for suggesting relevant baselines for our work. We discuss the comparison with two of the proposed works below:
>
> 1. **Bansal et al.**
> We agree that the work in [1] is relevant to our work, as both works present methods for end-to-end reasoning for a given set of problems, with the potential to be applied to larger instances. For the sake of comparison, we have trained a model for the N-Queens problem using the method in [1]. We provide a discussion of the limitations of the method compared to ours below:
> 	- **Uniqueness:** While we were able to successfully solve the 8-queens problem using the approach in [1], notice that the method, as it is, is purely deterministic, meaning that given the same input, it will always output the same solution. In the case of N-queens, where there are multiple solutions, and no input is provided, this is a limitation. To provide a comparison, while we can generate a correct solution using the method in [1] (DeepT), our approach can generate 37 distinct solutions out of 100 sampled solutions:
> 	| Model              	| Correct Instances       	| Unique Solutions          	|
> 	| -----------------------| ----------------------------| ------------------------------|
> 	| DeepT              	| 1                       	| 1                         	|
> 	| EBM (P=128) (ours) 	| 100                     	| 37                        	|
>
> 	- **Increasing computation:** while the method of [1] has the potential to solve larger instances by performing multiple iterative steps, we observe that increasing the number of steps from 30 (used during training) to 1000 led to a degeneracy in the solutions, which suggests that the model has learned suboptimal heuristics:
> 	| Num. Iterations | Size (Queens placed) |
> 	| ----------------| ----------------|
> 	| 30          	| 8           	|
> 	| 60          	| 7           	|
> 	| 100         	| 6           	|
> 	| 500         	| 6           	|
> 	| 1000        	| 6           	|
>
>       We will add these additional results to the appendix of the paper.
>
> 2. **Bevilacqua et al.**
> The work in [2] is closely related to ours, since both approaches try to learn iterative steps to solve a given problem and generalize to potentially larger problems. The approach to algorithmic reasoning presented in [2], however, proposes to supervise the learning of reasoning by directly supervising each algorithmic step, such that the model learns to execute a given algorithm. In other words, training requires access to a complete trace of the algorithm which the model learns to simulate. In contrast, our method relies only on input-output pairs (that is, no intermediate steps) and the reasoning is done through energy minimization, which makes both methods not directly comparable.
>
> ---
>
> ### References
>
> [1] Bansal et al., End-to-end Algorithm Synthesis with Recurrent Networks: Logical Extrapolation Without Overthinking
>
> [2] Bevilacqua et al., Neural Algorithmic Reasoning with Causal Regularisation

---

> ### Author Response · Authors · 2025-08-06
> **Feedback on Rebuttal**
>
> Dear reviewer xJaU,
>
> Thank you for the time and effort taken to review our paper and for providing valuable critique.
> As the discussion period is nearing the end, we would greatly appreciate it if you could kindly take a moment to review our response.
> We hope we have addressed the concerns raised regarding the need for specialised models and limited baseline comparisons.
> To this end, we have included novel results comparing our method against reasoning LLMs and suggested baselines.
> To further strengthen the evaluation of baselines from the literature, we include in the table below a comparison with the work from [3], as suggested by the reviewer:
>
> | Distribution     | Nodes      | XLVIN-CP             | EBM (Ours) (P=128)  |
> | ---------------- | ---------- | -------------------- | ------------------- |
> | Erdos Renyi      | [20, 39]  | $25.00 \pm 7.81$   | $8.60 \pm 4.82$   |
> |                  | [81, 99]  | $93.80 \pm 31.12$  | $29.20 \pm 8.05$  |
> | Holme Kim        | [22, 34]  | $29.00 \pm 7.75$   | $10.60 \pm 2.70$  |
> |                  | [86, 100] | $182.60 \pm 24.73$ | $59.00 \pm 3.74$  |
> | Regular Expander | [21, 40]  | $29.00 \pm 7.75$   | $11.00 \pm 4.89$  |
> |                  | [86, 100] | $112.60 \pm 10.97$ | $37.20 \pm 4.71$  |
> | Paley            | [19, 37]  | $151.80 \pm 92.13$ | $34.80 \pm 20.27$ |
> | Complete         | [8, 12]   | $34.80 \pm 16.42$  | $3.40 \pm 1.14$   |
>
> While the algorithmic bottleneck of the neural algorithmic reasoners
> is an interesting element, we want to note that
> the encoder element in [3] in the case of the graph coloring problem
> is still a GNN. As a result, similar to the GCN and GAT examples
> we provide in the paper, the model is able to learn
> to solve smaller in-distribution instances, but performance
> degrades significantly when applied to larger out-of-distribution instances.
> We hope this additional comparison helps address the reviewer's initial concerns.
> Please let us know if you have any additional questions or concerns.
>
> Thank you,
> Paper Authors
>
> --- References ---
> [3] Deac et al. Neural Algorithmic Reasoners are Implicit Planners, 2021

---

> > ### Comment · Reviewer_xJaU · 2025-08-08
> > **Nice Rebuttal**
> >
> > Thanks for all the clarity and the new results. My concerns are largely met and I'm increasing my score accordingly.

---

> > > ### Author Response · Authors · 2025-08-08
> > >
> > > We thank the reviewer for their time and positive answer. We are also glad that their concerns are largely met.
> > > We will update the manuscript accordingly to include the new results.

---

### Official Review · Reviewer_nBoZ · 2025-07-03

**Clarity:** 3
**Significance:** 3
**Originality:** 2
**Rating:** 5
**Confidence:** 3

**Summary:**

This paper introduces a novel approach to improving generalization in machine learning models for reasoning tasks by leveraging compositional energy minimization. The authors propose learning energy landscapes over subproblems and combining them during inference to tackle more complex problems. They introduce Parallel Energy Minimization (PEM) to optimize these composed energy functions effectively. The method is evaluated on reasoning tasks like N-Queens, 3-SAT, and Graph Coloring, demonstrating superior performance compared to existing state-of-the-art approaches.

**Questions:**

1. If the number of particles in PEM continues to increase (e.g. beyond 1024), will this persistently lead to better results? Additionally, for the EBM results in Table 8 of the main paper, would using more particles than 128 lead to better performance, particularly on the Holme-Kim graph distribution?
2. It seems the contrastive loss is important. How to tune the hyper-parameters of contrastive weight and MSE weight?
3. Based on the experimental settings in appendix, the total number of steps used for gradient descent is set to 100. Does the total timesteps parameter affect the optimization results?
4. Typo: the generation formulas for the positive and negative samples y+ and y- defined in line 128 are the same.

**Ethical Concerns:**

["NO or VERY MINOR ethics concerns only"]

**Final Justification:**

The authors provided a detailed response and additional analysis, including new experimental results on the crossword dataset, which addressed the initial concerns about generalization and applicability. Therefore, I raise my rating to 5.

**Limitations:**

yes

**Quality:**

3

**Strengths And Weaknesses:**

Strengths：
1. The paper is well‑written, with clear figures.
2. By decomposing complex reasoning problems into simpler subproblems, the approach aligns well with human-like reasoning processes.
3. The paper provides experimental validation across multiple reasoning tasks, showcasing significant improvements over baseline methods.

Weaknesses:
1. While PEM improves optimization, the requirement for multiple particles and iterations may increase computational overhead. A discussion on trade-offs between performance and computational resources would be beneficial.
2. The evaluation is limited to combinatorial problems which can be cleanly decomposed into subproblems. How to extend it to more complex reasoning tasks？

---

> ### Author Rebuttal · Authors · 2025-07-31
>
> We thank the reviewer for appreciating the clarity of our paper and the experimental validation. We believe that the feedback provided will help us improve the paper. We address the concerns below:
>
> ---
>
> ### **Q1) Increasing number of particles**
>
> We agree that we should have provided additional evaluation on the effect of increasing the number of particles. We have scaled the experiments from Table 8, particularly those from the Holme-Kim graph distribution, up to our hardware and time limits. We report the results in the table below. In this table, we show that increasing the number of particles does lead to an improvement in the number of conflicting edges, from 59.00 on average to 55.57 in the case of the larger instances. Notice, however, that we double the number of particles for each entry of the table to obtain improved results. That is, while there is some potential for further improvement, one needs to take into account the computational cost of increasing the number of particles. We will add this additional information to the appendix of the paper.
>
> | Num. Particles | Conflicting Edges (Holme Kim [22-34 nodes]) | Conflicting Edges (Holme Kim [86-100 nodes]) |
> |---------------| -------------------------------------------- | --------------------------------------------- |
> | 128        	| $10.60 \pm 2.70$          	| $59.00 \pm 5.24$
> | 256        	| $8.38 \pm 3.28$           	| $56.89 \pm 12.75$ |
> | 512        	| $9.68 \pm 3.13$           	| $54.68 \pm 8.10$  |
> | 1024       	| $7.04 \pm 3.64$           	| $57.04 \pm 3.27$  |
> | 2048       	| $7.79 \pm 4.20$           	| $55.57 \pm 4.58$  |
>
> ---
>
> ### **Q2) Loss weight tuning**
>
> Loss weight hyperparameters can indeed affect the end performance. In all our experiments, we used a weight of 1.0 for MSE and 0.5 for contrastive loss, since we observed that this combination led to the best performance in all cases. In early experiments, we also tried a weight of 1.0 for both losses without observing a significant difference in performance. Generally, we recommend values for the contrastive loss in the range 0.05-0.5. A higher weight for the contrastive loss might lead to a more unstable optimization, while a lower weight might render this loss ineffective.
>
> ---
>
> ### **Q3) Effect of number of timesteps**
>
> The optimization results are potentially affected by the number of timesteps, since more timesteps allow a smoother transition between states and enable more update steps. To evaluate the impact of this hyperparameter, we have trained a graph coloring model for timesteps in the range $[20, 50, 100, 200, 500, 1000]$. Below, we report the number of conflicting edges found by the model using the Holme-Kim random graph distribution. Each value is averaged over five random instances:
>
> | Num. Timesteps | Conflicting Edges (Holme-Kim \[22–34 nodes]) | Conflicting Edges (Holme-Kim \[86–100 nodes]) |
> | -------------- | -------------------------------------------- | --------------------------------------------- |
> | 20         	| $6.98 \pm 2.19$                          	| $44.27 \pm 7.29$                          	|
> | 50         	| $7.01 \pm 2.68$                          	| $54.40 \pm 8.61$                          	|
> | 100        	| $6.71 \pm 2.60$                          	| $52.97 \pm 8.07$                          	|
> | 200        	| $6.00 \pm 3.83$                          	| $50.31 \pm 5.76$                          	|
> | 500        	| $6.50 \pm 2.58$                         	| $49.63 \pm 3.19$                          	|
> | 1000       	| $6.51 \pm 2.35$                          	| $42.71 \pm 3.93$                             |
>
> Generally, we observe an improvement on average in the optimality of the solutions, going from 6.98 conflicting edges for 20 timesteps to 6.51 conflicting edges for 1000 timesteps in the smaller instances. In the case of the larger instances, we observe that with a number of timesteps equal to 20, the model does especially well, finding on average 44.27 conflicting edges. However, for the rest of the timesteps, the same trend is observed as in the smaller instances, with the number of conflicting edges decreasing from 54.40 to 42.71 as the number of timesteps increases from 50 to 1000. We can observe from this that the number of timesteps is a hyperparameter that, depending on the computational budget, can be tuned to improve the performance of the model. We will add these additional results to the appendix of the paper.
>
> ---
>
> ### **Q4) Typo in contrastive formulas**
>
> Thank you for pointing this out. We will rewrite the formulas, differentiating the positive and negative samples in line 128. The correct formulas are:
>
> $\tilde{y}^{+} = \sqrt{1 - \sigma_t} y^+ + \sigma_t \epsilon$ and $\tilde{y}^{-} = \sqrt{1 - \sigma_t} y^- + \sigma_t \epsilon$, where $\tilde{y}^{+}$ and $\tilde{y}^{-}$ are noise-corrupted versions of the positive and negative samples $y^+$ and $y^-$, respectively.

---

> > ### Comment · Reviewer_nBoZ · 2025-08-07
> >
> > Thanks to the authors for this additional analysis. I appreciate the thorough response and the additional experimental results on the crossword dataset, and I will raise my rating accordingly.

---

> > > ### Author Response · Authors · 2025-08-08
> > >
> > > We thank the reviewer for their time and for appreciating our responses.
> > > We are also glad that the additional results on the crossword dataset were helpful.
> > > We will update the paper accordingly to include the new results and comparisons.

---

> ### Author Response · Authors · 2025-08-06
> **Feedback on Rebuttal**
>
> Dear Reviewer nBoZ,
>
> Thank you for the time and effort taken to review our paper and providing valuable feedback.
> As the discussion period is nearing the end, we would greatly appreciate it if you could kindly take a moment to review our response.
> We hope that our answer has helped clarify the impact of different hyperparameters on the performance of our method.
> In other reviewers' responses, we have also addressed additional concerns regarding the trade-off between computational cost
> and performance, as well as application to other tasks.
> Please, let us know if you have any additional questions or concerns.
>
> Thank you,
> Paper Authors

---

### Decision · Program_Chairs · 2025-09-17

**Decision:**

Accept (spotlight)

**Comment:**

This paper introduces a novel and intuitive method for improving generalization on reasoning tasks by composing learned energy landscapes of smaller subproblems. The reviewers were in strong agreement about the quality and contributions of the work.

Summary of Strengths:

* The proposed approach is novel, well-motivated, and aligns well with compositional reasoning (nBoZ, 6UFB, 7Yz8).

* The paper is well-written, clear, and easy to follow (nBoZ, 6UFB, 7Yz8).

* The method demonstrates strong empirical performance, showing significant improvements over state-of-the-art methods on a variety of combinatorial tasks (nBoZ, 6UFB, 7Yz8).

* The authors provided a thorough rebuttal with new experiments that addressed nearly all reviewer concerns, including new baselines (xJaU), application to a new domain (crosswords, per 7Yz8's suggestion), and analysis of computational trade-offs (nBoZ, 6UFB).

The reviewers unanimously recommend acceptance. The core idea is technically solid, and the experimental results are impressive. Initial concerns regarding the scope of evaluation, comparison to modern baselines like LLMs, and computational cost were effectively addressed by the authors in their detailed rebuttal, leading all reviewers to either confirm or raise their scores. This paper presents a valuable contribution to the field of reasoning and generalization.